# ATTIC: A NEW ARCHITECTURE FOR TABULAR IN-CONTEXT LEARNING TRANSFORMERS

## ABSTRACT

Tabular In-Context Learning (ICL) transformers, such as TabPFN and Tab-ForestPFN, have shown strong performance on tabular classification tasks. In this paper, we introduce Attic, a new architecture for ICL-transformers. Unlike TabPFN and TabForestPFN, where one token represents all features of one observation, Attic assigns one token to each feature of every observation. This simple architectural change results in a significant performance boost. As a result, we can confidently say that neural networks outperform tree-based methods like XG-Boost.

## 1 INTRODUCTION

Tabular classification is an important prediction task in many different parts of industry. This task concerns predicting the value of a certain feature for an observation, given other features, often stored in tabular format. Tabular classification can be used to predict heart disease based on patient characteristics (Singh & Kumar, 2020), to predict whether someone commits credit card fraud (Awoyemi et al., 2017), or to predict the chance an employee quits the company (Fallucchi et al., 2020). Numerous methods from wildly different backgrounds have been used to tackle this task, with the most popular being tree-based methods like XGBoost (Chen & Guestrin, 2016) or ensemble methods like AutoGluon (Erickson et al., 2020).

Recently, there has been work that tries to improve the classification performance by pretraining an In-Context Learning (ICL) transformer. ICL-transformers like TabPFN (Hollmann et al., 2023) perform zero-shot inference after pretraining on synthetic data, while TabForestPFN (Breejen et al., 2024) further fine-tunes this ICL-transformer for better performance. Both use the same transformer architecture, where one token represents all the features of one observation. We refer to this as an 'observation token'. Using observation tokens, these ICL-transformers embed the features of one observation into a token via a linear embedding.

This observation token constrains the ICL-transformer in two ways. First of all, the embedding layer has a maximum size; TabPFN and TabForestPFN both accept a maximum of a hundred features. Secondly, these ICL-transformers are dependent on the order of the features. In tabular data, the order of the columns should not influence the final prediction, but in ICL-transformers like TabPFN and TabForestPFN, the linear embedding layer enforces an arbitrary order in the features.

In this paper, we introduce a straightforward modification to the architecture: replacing the observation token with a cell token, which represents a single feature of each observation. This enables the ICL transformer to accommodate as many features as the GPU can handle while maintaining feature-order invariance. We refer to this modified ICL transformer as Attic: "A Tabular Transformer based on In-Context Learning." Given this architectural change, the memory required to fit everything on the GPU increases significantly. We tackle this issue by using FlashAttention and bfloat16 mixed-precision.

Not only does this architectural change allow the network to use an arbitrary number of features, but we also observe an immense performance increase. The performance increase is so significant that it outperforms XGBoost and CatBoost on average on two benchmarks (Grinsztajn et al., 2022; McElfresh et al., 2023) even when these methods are allowed to run hundreds of hyperparameter searches, while Attic does not run any hyperparameter search at all. Especially for datasets with

more than 500 observations, there are very few datasets in the benchmark suite for which XGBoost or CatBoost have better performance.

Additionally, we found a dataset for which Attic has 20% higher accuracy than any other method, including ensemble methods like AutoGluon. This shows that Attic is a strong method that should be included in ensemble methods because it can achieve performance levels on some datasets that no other methods can attain. Given these strong results, we are excited to see future developments in the field of tabular ICL.

## 2 RELATED WORK

The most popular methods for tabular classification are tree-based methods such as XGBoost (Chen & Guestrin, 2016), LightGBM (Ke et al., 2017), and CatBoost (Prokhorenkova et al., 2018). The fundamental principle behind tree-based methods is to iteratively split the feature space and assign predictions based on the resulting partitions. Recent tabular prediction benchmarks (Gorishniy et al., 2021; Grinsztajn et al., 2022; McElfresh et al., 2023; Zabërgja et al., 2024) show that these tree-based methods perform exceptionally well.

Numerous studies have tackled tabular data using neural networks (Kadra et al., 2021; Somepalli et al., 2021; Gorishniy et al., 2022). Generally, their performance is lacking compared to tree-based methods, although progress is still being made (Gorishniy et al., 2024). Neural networks trained from scratch seem to struggle in low-data regimes because they may lack specific biases that tree-based methods have (Grinsztajn et al., 2022).

Other research has focused on using language data for their predictions (Yang et al., 2023; Kim et al., 2024; Yan et al., 2024). When datasets are small, pretrained language models can infer relationships between features based on feature names and table metadata. This capability allows such approaches to achieve excellent performance in a few-shot setting (Hegselmann et al., 2023; Gardner et al., 2024). However, these methods are unlikely to scale effectively to larger datasets and are heavily dependent on having informative feature names, which are not present in all datasets.

Our work contributes to the field of tabular In-Context Learning (ICL)(Hollmann et al., 2023). In this approach, transformers are pretrained to predict test observations using training observations provided in the context. A major limitation of ICL-transformers is the necessity of including the entire training dataset within the context, so there is active research on how to address this issue (Ma et al., 2023; Feuer et al., 2024; Breejen et al., 2024; Thomas et al., 2024)

Regarding neural network architectures, transformers trained from scratch use various strategies. SAINT (Somepalli et al., 2021) uses cell tokens, whereas FT-Transformer (Gorishniy et al., 2021) uses feature tokens. Benchmark results (Grinsztajn et al., 2022; McElfresh et al., 2023) indicate no clear relationship between the token scheme and performance. Additionally, we have not identified any literature suggesting that cell tokens are superior to observation tokens for modeling tabular data. This gap shows the relevance of our work, which specifically compares token schemes for ICL-transformers.

## 3 METHODOLOGY

In tabular classification, we are interested in predicting labels $\boldsymbol{y} \in \mathbb{N}^n$ given features $\boldsymbol{X} \in \mathbb{R}^{n \times k}$, where $k$ is the number of features and $n$ is the number of observations. In tabular In-Context Learning (ICL), we consider a support set $(\boldsymbol{X}_S, \boldsymbol{y}_S)$, where $\boldsymbol{y}_S$ is known, and a query set $(\boldsymbol{X}_Q, \boldsymbol{y}_Q)$, where $\boldsymbol{y}_Q$ must be predicted or is used as a loss during training. During inference, it is natural to think of the support set as a training dataset and the query set as the test dataset. However, this comparison breaks down during pretraining and fine-tuning, as the query set then also comes from the training dataset. An ICL-transformer takes $(\boldsymbol{X}_S, \boldsymbol{y}_S, \boldsymbol{X}_Q)$ as input and predicts $\boldsymbol{y}_Q$.

We train Attic using the same tabular ICL-pipeline as described by Breejen et al. (2024). This means that during pretraining, new datasets are generated at each step using a synthetic dataset generator. This generator combines the the TabPFN dataset generator (Hollmann et al., 2023) with the forest dataset generator (Breejen et al., 2024). The generated datasets are preprocessed and split into support and query sets before being fed into the ICL-transformer. During fine-tuning, the

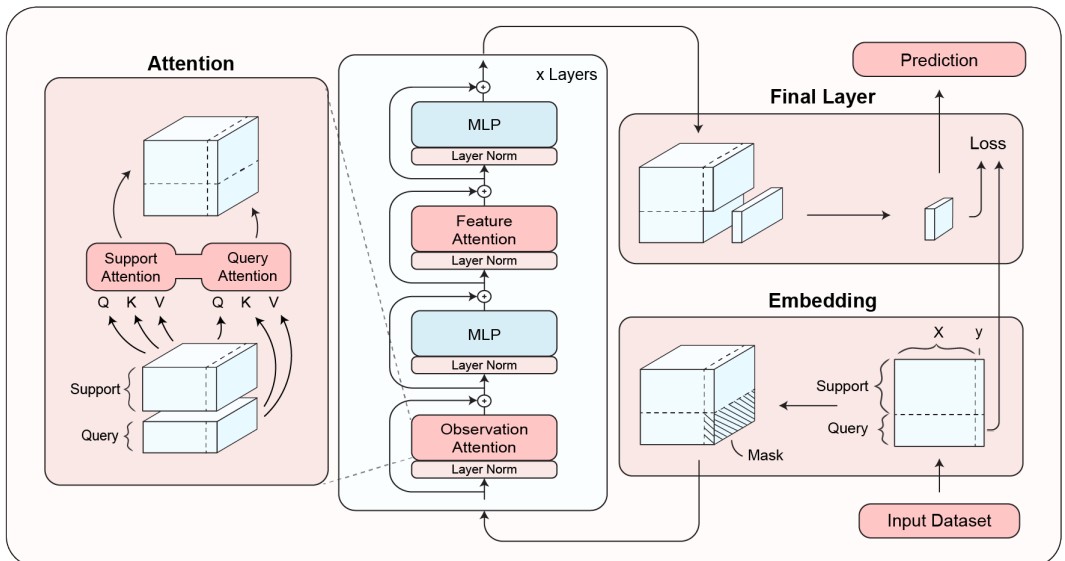

Figure 1: Architecture of Attic

ICL-transformer is further trained by drawing support and query sets from the training dataset of the specific real-world dataset being used. Inference is done by selecting the support set from the training dataset and the query set from the validation or test dataset.

## 3.1 ARCHITECTURE

In this paper, we introduce Attic, an ICL-transformer based on cell tokens. The architecture is depicted in Figure 1. The core difference compared to TabPFN (Hollmann et al., 2023) and Tab-ForestPFN (Breejen et al., 2024) is what a token represents. Attic uses cell tokens, meaning that one token represents one feature of one observation, as opposed to TabPFN's observation token, which represents all features of one observation.

Given a value $x_{ij} \in \mathbb{R}$ of observation $i$ and feature $j$, Attic embeds this value $x_{ij}$ into token $h_{ij} \in \mathbb{R}^d$, where $d$ is the dimension of the model. These tokens pass through $L$ layers, each consisting of an observation attention layer, a feature attention layer, two MLP layers, and layer normalizations before each module. The MLP layers consist of two linear layers, with the inner dimension being four times larger than $d$, and a GeLU activation in between. The final layer of the model isolates $y_Q$ from the other tokens and maps it to the number of classes, which is fixed at 10 classes in all ICL-transformers discussed.

The observation attention treats the observation dimension as the sequence and the feature dimension as a batch dimension, while feature attention does the opposite. The observation attention mechanism uses a mask that ensures that observations from the query set cannot see other observations from the query set. This guarantees that the prediction for a test observation remains independent of the other test observations in the context. This attention mechanism is borrowed from TabPFN.

Compared to TabPFN and equivalently TabForestPFN, the tokens of Attic have an additional feature dimension, which increases the total memory requirement by a factor $k$. The embedding is also different, since the TabPFN architecture embeds from observation $x_i \in \mathbb{R}^k$ to token $h_i \in \mathbb{R}^d$. Our embedding also embeds the labels of $y_S$ as if it were a language tokens, with each class treated as a word, in contrast to TabPFN, which treats $y_S$ as a float vector. We changed this because we believe this formulation is more natural, but performance-wise, it has little impact. Other than the differences mentioned, there are no additional architectural changes between Attic and TabPFN.

Table 1: Model comparison overview.

| | TabForestPFN | | | Attic |
|---|---|---|---|---|
| | Original | BF16-M | BF16-L | |
| **Architecture** | | | | |
| Based on observation tokens | ✔ | ✔ | ✔ | ✘ |
| Based on cell tokens | ✘ | ✘ | ✘ | ✔ |
| Uses FlashAttention | ✘ | ✔ | ✔ | ✔ |
| Uses mixed-precision with bfloat16 | ✘ | ✔ | ✔ | ✔ |
| Hidden dimension size | 512 | 512 | 2048 | 512 |
| Number of layers | 12 | 12 | 24 | 12 |
| **Hyperparameters** | | | | |
| Batch size | 512 | 1024 | 1024 | 1024 |
| Number of steps | 50k | 50k | 50k | 44k |
| Number of generated pretraining datasets | 26M | 51M | 51M | 45M |
| Learning rate | 1e-4 | 1e-4 | 1e-4 | 1e-3 |
| Weight decay | 0.0 | 0.0 | 0.0 | 0.1 |
| Pretraining dataset minimum number of observations | 128 | 128 | 128 | 16 |
| Pretraining dataset maximum number of observations | 1024 | 1024 | 1024 | 512 |
| Pretraining dataset minimum number of features | 3 | 3 | 3 | 1 |
| Pretraining dataset maximum number of features | 100 | 100 | 100 | 16 |
| **Statistics** | | | | |
| Training time (H100 GPU-hours) | 199 | 97 | 299 | 208 |
| Parameter count | 39M | 39M | 1226M | 76M |

## 3.2 MOTIVATION

In the results, we will see that this cell-token architecture is significantly more performant than the observation-token architecture. In our intuition, this is because the cell-token architecture is feature-order invariant. For tabular data, the order of the features holds no importance: in an Excel sheet, you can freely rearrange the columns, and this should not influence the prediction. However, observation-token-based ICL-transformers do enforce a specific feature order; the linear mapping between features and tokens changes when the features are reordered.

We believe that this dependency on feature order leads to training inefficiencies. The cell-token ICL-transformer treats each feature the same and learns how to construct relationships between features. In contrast, the observation-token ICL-transformer assigns each feature a unique position in the embedding. Consequently, when learning the relationships between features, it has to learn this relationship for every possible position that these features can be in.

## 4 EXPERIMENTS

Our experiments focus on showing the improvement of Attic over TabForestPFN (Breejen et al., 2024). In Section 4.1, we describe the design choices for the comparison, conducted on the benchmarks outlined in Section 4.2. Section 4.3 presents the main results, while Sections 4.5, 4.6, and 4.7 delve deeper into the findings. Finally, we include some initial regression results in Section 4.8.

## 4.1 DESIGN

For the architectural comparison between Attic and TabForestPFN, we want to change the architecture while keeping all other hyperparameters the same. However, the computational costs of Attic scale with both the number of features and the number of observations, in contrast to TabForestPFN, which scales only with the number of observations. For this reason, we pretrain Attic using smaller pretraining datasets. Furthermore, Attic uses FlashAttention (Dao et al., 2022) and mixed-precision with bfloat16, so it is important to implement these techniques on TabForestPFN for a fair comparison.

Table 1 reports all the differences between Attic and TabForestPFN, including the variants BF16-M and BF16-L, which implement mixed precision and FlashAttention for two different model sizes. The number of training steps for Attic has been set to ensure that Attic trains for approximately 200 GPU-hours on an H100, matching the training time of TabForestPFN. With TabForestPFN BF16-L, we can assess the sample efficiency of Attic, as TabForestPFN BF16-L is a larger model trained on a greater number of generated pretraining datasets. We include TabForestPFN BF16-M to show the effect of mixed-precision training on TabForestPFN.

Other differences in hyperparameters do not affect the fairness of the comparison. Optimizer hyperparameters such as the learning rate and weight decay are model-specific; we experienced training collapse when using these settings on TabForestPFN. Additionally, pretraining Attic on smaller datasets than TabForestPFN favors TabForestPFN, as almost all our benchmark datasets contain more than 512 observations or more than 16 features. This makes it more challenging for Attic to generalize to larger datasets. Given this experimental design, if Attic outperforms TabForestPFN and its variants, we can conclude that Attic has a better architecture.

## 4.2 BENCHMARKS

We evaluate the ICL-transformers on two benchmarks: the benchmark we refer to as WhyTrees (Grinsztajn et al., 2022) and TabZilla (McElfresh et al., 2023). For most of the methods, we rely on publicly available results, which we further extend by running TabPFN, TabForestPFN, AutoGluon (Erickson et al., 2020), and Attic ourselves.

WhyTrees is a benchmark that tests on datasets ranging from 1,000 to 10,000 observations, providing a total of 25 classification datasets. These datasets are categorized into 'numerical' datasets, which include only numerical features, and 'mixed' datasets, which also includes categorical features. The benchmark authors run methods with up to a few hundred hyperparameter search iterations.

TabZilla is a benchmark consisting of 176 datasets, of which we test on 94. See Appendix A.2 for details on the dataset selection. The datasets in TabZilla vary in size from 10 to 100,000 observations and include between 2 to hundreds of features. Hyperparameter searches for the methods are conducted up to 30 iterations.

When running TabPFN, TabForestPFN, and Attic on these benchmarks, we run each method ten times and report the average performance across these runs. AutoGluon, using the 'best quality' setting, is only run once on the WhyTrees benchmark and is not run on TabZilla due to extremely long running times. The run times for these methods can be found in Appendix A.4, and metadata about the datasets for both benchmarks is available in Appendix A.3.

## 4.3 MAIN RESULTS

Figure 2 reports the main results of Attic on the WhyTrees benchmark compared to other baselines provided by the benchmark. Table 3 shows the results of Attic against other ICL-transformers and AutoGluon. The results for the TabZilla benchmark are presented in Table 2.

First, we compare Attic with the TabForestPFN variants. Both benchmarks indicate that Attic outperforms all variants of TabForestPFN by a wide margin. A closer examination of the results reveals that switching to bfloat16 severely diminishes the performance of TabForestPFN, and that scaling up this model barely recovers the original performance. This shows that Attic's superior performance cannot be attributed to the efficiency improvements from using FlashAttention (Dao et al., 2022) combined with bfloat16 mixed-precision. Considering the other arguments discussed in Section 4.1, we conclude that the excellent performance of Attic stems from the architectural changes.

Looking at other baselines, fine-tuned Attic outperforms all other methods on both the WhyTrees and the TabZilla benchmarks. This is a strong result, especially since Attic does not use any hyperparameter optimization, wheras other methods perform extensive hyperparameter sweeps. To further put the results in perspective, we included AutoGluon as an additional reference. AutoGluon is a method that runs hundreds of tree-based algorithms, neural networks, and other tabular prediction models, then ensembles them. As Attic can be included in this ensemble, we do not consider it a direct competitor; instead, it highlights how strong Attic is.

Table 2: Main Results for the TabZilla benchmark. N. Accuracy stands for Normalized accuracy. Rank compares the relative rank of a method compared to all other methods on that dataset.

| Models | Rank | | | | N. Accuracy | |
|---|---|---|---|---|---|---|
| | min | max | mean | median | mean | median |
| Attic - Fine-tuned | 1 | 28 | **6.4** | **4.0** | **0.882** | **0.961** |
| Attic - Zero-shot | 1 | 26 | 8.8 | 6.2 | 0.834 | 0.922 |
| TabForestPFN - Fine-tuned | 1 | 28 | 9.2 | 7.0 | 0.829 | 0.886 |
| CatBoost | 1 | 24 | 10.4 | 10.0 | 0.832 | 0.861 |
| TabForestPFN - Zero-shot | 1 | 25 | 10.4 | 10.0 | 0.809 | 0.880 |
| TabPFN - Fine-tuned | 1 | 27 | 10.5 | 10.5 | 0.823 | 0.875 |
| XGBoost | 1 | 25 | 10.7 | 11.0 | 0.826 | 0.877 |
| TabForestPFN BF16-L - Fine-tuned | 1 | 27 | 12.2 | 11.0 | 0.785 | 0.868 |
| TabForestPFN BF16-M - Fine-tuned | 1 | 26 | 12.5 | 12.2 | 0.784 | 0.850 |
| LightGBM | 2 | 28 | 12.5 | 12.2 | 0.777 | 0.860 |
| TabForestPFN BF16-L - Zero-shot | 1 | 26 | 12.8 | 12.2 | 0.770 | 0.852 |
| RandomForest | 1 | 27 | 12.9 | 12.5 | 0.782 | 0.835 |
| TabPFN - Zero-shot | 1 | 27 | 13.0 | 13.0 | 0.767 | 0.830 |
| Resnet | 1 | 28 | 13.5 | 13.0 | 0.719 | 0.834 |
| SAINT | 1 | 28 | 13.8 | 14.0 | 0.721 | 0.795 |
| NODE | 2 | 28 | 13.8 | 14.5 | 0.741 | 0.817 |
| SVM | 1 | 27 | 14.1 | 14.5 | 0.701 | 0.798 |
| FT-Transformer | 1 | 25 | 14.4 | 15.0 | 0.724 | 0.794 |
| DANet | 3 | 28 | 16.4 | 16.0 | 0.708 | 0.757 |
| TabForestPFN BF16-M - Zero-shot | 1 | 27 | 17.1 | 17.0 | 0.708 | 0.795 |
| MLP-rtdl | 1 | 28 | 17.8 | 19.8 | 0.613 | 0.723 |
| STG | 2 | 28 | 18.0 | 19.8 | 0.585 | 0.672 |
| LinearRegression | 1 | 28 | 19.4 | 22.0 | 0.559 | 0.590 |
| MLP | 2 | 28 | 19.6 | 22.0 | 0.563 | 0.582 |
| TabNet | 3 | 28 | 19.9 | 21.0 | 0.571 | 0.661 |
| DecisionTree | 1 | 28 | 20.6 | 22.0 | 0.496 | 0.551 |
| KNN | 2 | 28 | 21.4 | 24.0 | 0.467 | 0.478 |
| VIME | 2 | 28 | 23.7 | 26.0 | 0.337 | 0.238 |

When examining the individual datasets within WhyTrees in more detail, we find one dataset where the performance gap between Attic and all other methods is massive. Figure 3 illustrates the results on the Eye Movements dataset. Here, Attic outperforms all other methods, including AutoGluon, by more than 20%. This further suggests that Attic is an exceptionally strong model. Given that this specific dataset significantly influences the averaged normalized accuracy, we refer the reader to Appendix A.5 for detailed results on other individual datasets.

## 4.4 MIXED-PRECISION TRAINING

In the main results, we have seen that Attic outperforms all other TabForestPFN variants. In particular, there is a significant performance drop when switching the mixed-precision setup of TabForestPFN from float32 to bfloat16, as shown in Tables 2 and 3 when comparing the original TabForestPFN with variant BF16-M. This indicates that TabForestPFN is highly sensitive to floating-point precision. The exact cause of this performance deterioration is unknown to us. We would have liked to determine if Attic is also sensitive to precision. However, we cannot evaluate Attic using float32 because we cannot fit it into GPU memory without FlashAttention, which does not support float32.

Given this sensitivity to precision, it would be natural to try float16 instead of bfloat16. However, we encountered high pretraining instability with float16, where the cross-entropy training loss would start to climb and diverge at a seemingly random point during training. For weeks, we experimented with gradient scaler settings in an attempt to stabilize the training, but unfortunately, all pretraining runs eventually collapsed. This issue occurred for both TabForestPFN and Attic.

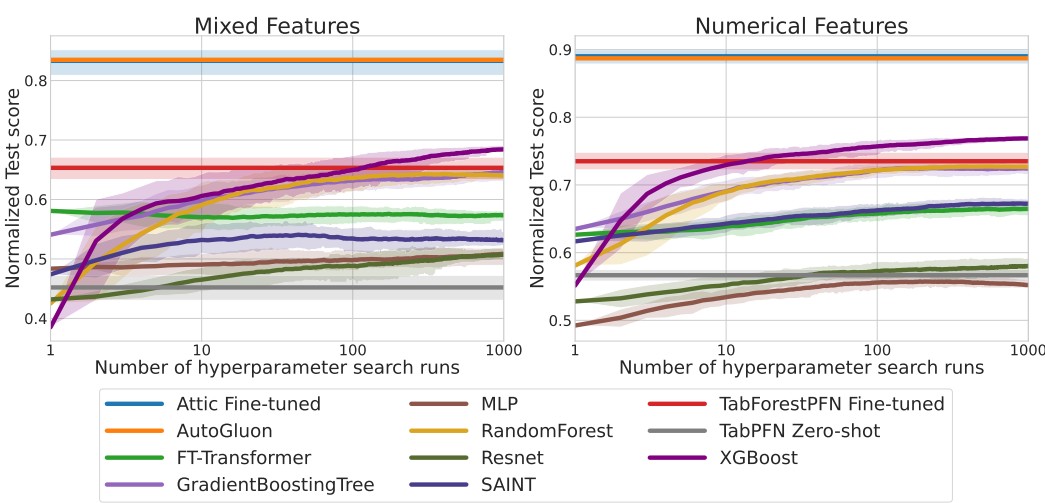

Figure 2: Main results on the WhyTrees Benchmark. ICL-transformers report the mean accuracy over ten default runs for different fine-tuning seeds, Autogluon reports one default run, and all others use random search over the hyperparameters. Results by dataset are displayed in Figure 7 and 8 of the appendix.

## 4.5 COMPUTATION RESOURCES

Run times, as reported in Appendix A.4, tell us that in the fine-tuning setting, Attic takes an average of 125 seconds to complete one cross-validation split. On average, Attic is twice as slow as Tab-ForestPFN, although this varies depending on the number of features. When dealing with datasets containing 5 or 10 features, the fine-tuning speed is similar. However, for datasets with 400 features, Attic experiences a slowdown by a factor of ten.

Attic also requires more GPU memory than TabForestPFN. As Attic uses cell tokens, it needs additional GPU memory to store all tokens for the backward pass during fine-tuning. With a maximum support size of 8192 and a maximum query size of 1024, evaluating TabForestPFN can be done within 32GB of memory, while Attic requires 80GB to run on all datasets in the benchmark. We considered increasing the context size of TabForestPFN to 16,384 for a fairer comparison, but this only marginally improved the prediction accuracy of TabForestPFN. This is because the maximum training dataset size in the WhyTrees benchmark is 10,000, and in Tabzilla there are only a handful of datasets larger than this.

## 4.6 DECISION BOUNDARIES

Given the strong results presented in the main results section, we now examine the behavior of the new architecture regarding decision boundaries. Breejen et al. (2024) demonstrated that ICL-transformers can create highly complex decision boundaries when fine-tuned. We reproduce their analysis and compare the decision boundaries of Attic and TabForestPFN.

Figure 4 shows the decision boundaries for TabForestPFN and Attic based on the two most important variables of the Electricity dataset. We observe that, on this dataset, Attic can create more detailed decision boundaries than TabForestPFN. The shape of Attic's decision boundaries is more similar to those of random forest than to those of TabForestPFN. Attic also provides a major boost in the accuracy compared to TabForestPFN. We attribute this behavior to the fact that cell tokens offer a better representation of tabular data than observation tokens (see Section 3.2 for further motivation).

## 4.7 INDIVIDUAL DATASETS

In Section 4.3, we have seen that Attic has excellent performance on the TabZilla benchmark. Here, we examine the results of individual TabZilla datasets in more detail to identify potential patterns.

Table 3: WhyTrees normalized accuracy results for methods without hyperparameter search.

|  | Mixed | Numerical |
|---|---|---|
| **Zero-shot** | | |
| TabPFN | **0.452** | 0.567 |
| TabForestPFN | 0.419 | 0.597 |
| TabForestPFN BF16-M | 0.291 | 0.385 |
| TabForestPFN BF16-L | 0.423 | 0.574 |
| Attic | 0.446 | **0.610** |
| **Fine-tuned** | | |
| TabPFN | 0.587 | 0.684 |
| TabForestPFN | 0.654 | 0.734 |
| TabForestPFN BF16-M | 0.538 | 0.613 |
| TabForestPFN BF16-L | 0.608 | 0.703 |
| Attic | **0.832** | **0.890** |
| **Other** | | |
| AutoGluon | 0.835 | 0.887 |

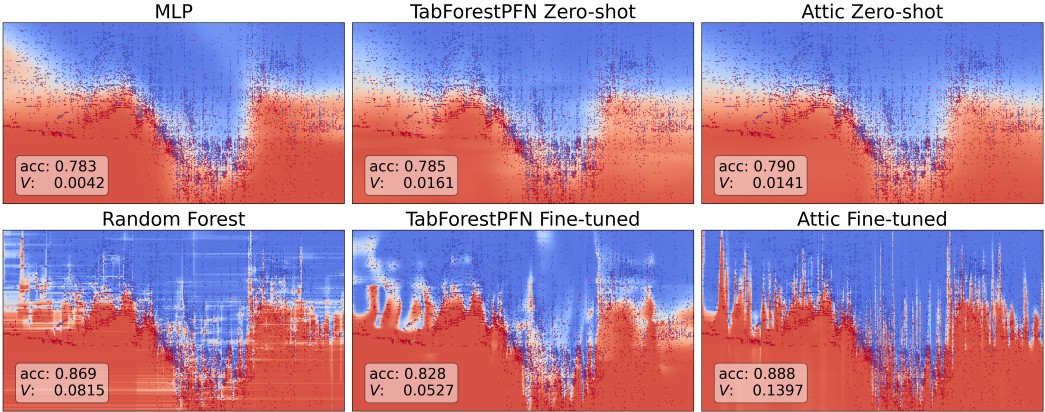

Figure 3: Results on the Eye Movements dataset (OpenML ID 44156). Other datasets are presented in Appendix A.5

Figure 4: Decision boundaries for the Electricity dataset (OpenML ID 44156). Axis represent features, colors are predicted class probabilities, and dots are test observations. Score $V$ measures complexity (see Appendix A.7).

Figure 5 presents comparisons of performance between fine-tuned Attic, CatBoost, XGBoost, fine-tuned TabForestPFN, and zero-shot Attic.

First, we compare Attic with TabForestPFN and observe that the performance gains are primarily seen in datasets with more than a thousand observations. When comparing Attic with CatBoost and XGBoost, we find that there are still a few datasets, particularly those with fewer than a thousand observations, where tree-based models perform better. This indicates that although Attic is generally stronger on average, there remains room for improvement.

When comparing Attic's fine-tuning and zero-shot performance, we observe a clear threshold around 500 observations. Zero-shot Attic performs better on datasets with fewer than 500 observations, while fine-tuned Attic excels on datasets with more than 500 observations. Since pretraining occurs on datasets with a maximum size of 512 observations, it is likely that fine-tuning is more effective for datasets larger than 512 observations because Attic has not encountered such large datasets during pretraining. Given the performance deficit of fine-tuned Attic on datasets with fewer than 500 observations, we include similar comparison graphs for zero-shot Attic in Appendix A.6.

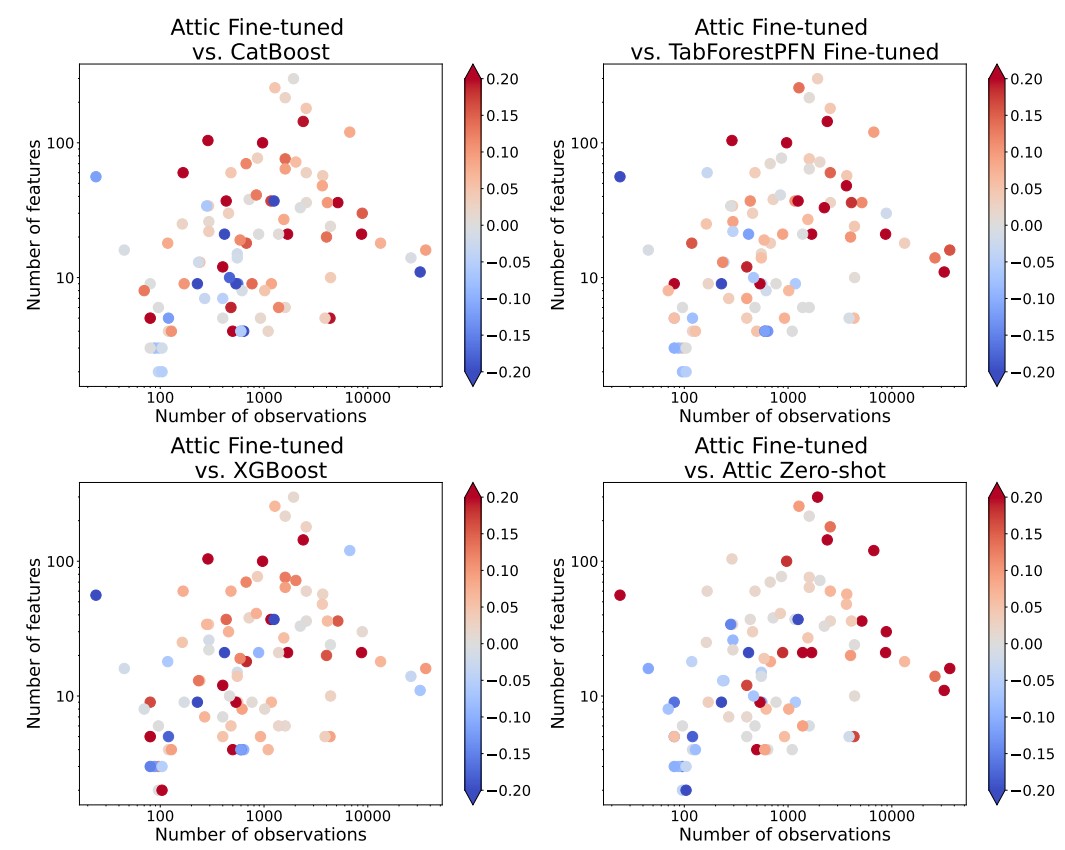

Figure 5: Comparison of fine-tuned Attic with XGBoost, CatBoost, TabForestPFN and zero-shot Attic. Dots represent difference in normalized accuracy for an individual dataset from TabZilla. Red means fine-tuned Attic is better.

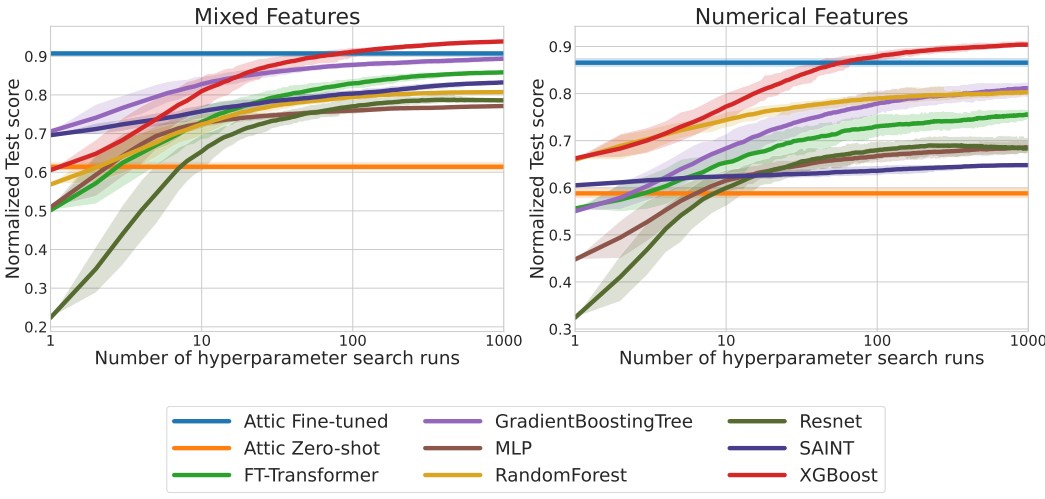

Figure 6: Regression results on the WhyTrees Benchmark. ICL-transformers report the mean R2 score over ten default runs for different fine-tuning seeds, and all others use random search over the hyperparameters. Results by dataset are displayed in Figure 9 and 10 of the appendix.

## 4.8 REGRESSION

In this section, we present initial experiments of Attic on regression tasks. To adapt Attic for regression, we modified the synthetic dataset generator to output regression tasks instead of classification tasks. Additionally, we replaced the embedding and final layer of the Attic with linear layers that enable the model to input and output float values instead of class labels. The loss function was changed from cross-entropy loss to mean-squared error.

During the pretraining of Attic under these settings, we experienced high training instability that was not present in the classification setting. To address this, we reduced the pretraining learning rate from 1e-3 to 1e-4 and the fine-tuning learning rate from 1e-5 to 1e-6. The results under these adjusted settings are presented in Figure 6.

The performance of Attic on regression tasks is not as impressive as its performance on classification tasks. We did not anticipate that regression would behave differently from classification, indicating that further investigation into regression tasks is necessary. Nonetheless, Attic remains the best-performing neural network method for regression, where we note that regression versions of TabPFN and TabForestPFN do not exist.

## 5 CONCLUSION

In this paper, we present Attic, an ICL-transformer that uses cell tokens instead of observation tokens. Our experiments show that this new architecture leads to large improvements in performance over TabForestPFN when trained under the same computational budget.

When comparing Attic with XGBoost and CatBoost, we see that Attic outperforms these tree-based methods on average, particularly on datasets with more than 500 observations. In this comparison, we fine-tuned Attic on default settings, while XGBoost and CatBoost were allowed to perform hyperparameter searches. The improvement over XGBoost and CatBoost suggests that we finally have surpassed a major barrier in AI for tabular data, favoring ICL-transformers over traditional tree-based methods.

The next milestone is to outperform AutoGluon. Until then, incorporating Attic into ensemble methods appears to be a promising approach. However, as we have observed that Attic still faces challenges with regression tasks, our immediate focus will be on improving performance in that area. Given that the field of tabular ICL-transformers is still in its early stages, we anticipate significant advancements in the coming years.

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

## A APPENDIX

### A.1 REPRODUCIBILITY STATEMENT

With this submission, we provide all the code that is necessary to pretrain, fine-tune and evaluate our model as well as other models on the benchmark datasets. We will also provide the exact scripts and hyperparameters settings necessary to reproduce the training and evaluation, as well as intermediate run statistics and outputs, and the code to reproduce all tables and graphs. Code will be hosted on GitHub and pretrained weights will be publicly available.

### A.2 TABZILLA DATASET SELECTION

From the TabZilla benchmark, we test on 94 datasets from a collection of 176. We follow the selection procedure as outlined by the authors of TabForestPFN (Breejen et al., 2024), which means we remove all datasets for which a baseline algorithm did not have a completed run in the results provided by the benchmark authors. Additionally datasets that have more than 10 classes are also removed.

### A.3 BENCHMARK METADATA

Tables 4 and 5 show the composition of all the datasets we use to benchmark Attic.

Table 4: Metadata of the WhyTrees Benchmark. Splits refers to the number of cross validation splits.

| OpenML | | Observations | | | | Features | Splits | Classes |
|---|---|---|---|---|---|---|---|---|
| ID | Name | All | Train | Valid | Test | | | |
| 44089 | credit | 16714 | 10000 | 2014 | 4700 | 10 | 2 | 2 |
| 44120 | electricity | 38474 | 10000 | 8542 | 19932 | 7 | 1 | 2 |
| 44121 | covertype | 566602 | 10000 | 50000 | 50000 | 10 | 1 | 2 |
| 44122 | pol | 10082 | 7057 | 907 | 2118 | 26 | 3 | 2 |
| 44123 | house_16H | 13488 | 9441 | 1214 | 2833 | 16 | 3 | 2 |
| 44125 | MagicTelescope | 13376 | 9363 | 1203 | 2810 | 10 | 3 | 2 |
| 44126 | bank-marketing | 10578 | 7404 | 952 | 2222 | 7 | 3 | 2 |
| 44128 | MiniBooNE | 72998 | 10000 | 18899 | 44099 | 50 | 1 | 2 |
| 44129 | Higgs | 940160 | 10000 | 50000 | 50000 | 24 | 1 | 2 |
| 44130 | eye_movements | 7608 | 5325 | 684 | 1599 | 20 | 3 | 2 |
| 44156 | electricity | 38474 | 10000 | 8542 | 19932 | 8 | 1 | 2 |
| 44157 | eye_movements | 7608 | 5325 | 684 | 1599 | 23 | 3 | 2 |
| 44159 | covertype | 423680 | 10000 | 50000 | 50000 | 54 | 1 | 2 |
| 45019 | Bioresponse | 3434 | 2403 | 309 | 722 | 419 | 5 | 2 |
| 45020 | default-of-cred... | 13272 | 9290 | 1194 | 2788 | 20 | 3 | 2 |
| 45021 | jannis | 57580 | 10000 | 14274 | 33306 | 54 | 1 | 2 |
| 45022 | Diabetes130US | 71090 | 10000 | 18327 | 42763 | 7 | 1 | 2 |
| 45026 | heloc | 10000 | 7000 | 900 | 2100 | 22 | 3 | 2 |
| 45028 | california | 20634 | 10000 | 3190 | 7444 | 8 | 1 | 2 |
| 45035 | albert | 58252 | 10000 | 14475 | 33777 | 31 | 1 | 2 |
| 45036 | default-of-cred... | 13272 | 9290 | 1194 | 2788 | 21 | 3 | 2 |
| 45038 | road-safety | 111762 | 10000 | 30528 | 50000 | 32 | 1 | 2 |
| 45039 | compas-two-year... | 4966 | 3476 | 447 | 1043 | 11 | 3 | 2 |

Table 5: Metadata of the TabZilla Benchmark. Splits refers to the number of cross validation splits.

| OpenML | | Observations | | | | Features | Splits | Classes |
|---|---|---|---|---|---|---|---|---|
| ID | Name | All | Train | Valid | Test | | | |
| 3 | kr-vs-kp | 3196 | 2556 | 320 | 320 | 36 | 10 | 2 |
| 4 | labor | 57 | 45 | 6 | 6 | 16 | 10 | 2 |

| | | | | | | | | |
|---|---|---|---|---|---|---|---|---|
| 9 | autos | 205 | 163 | 21 | 21 | 25 | 10 | 6 |
| 10 | lymph | 148 | 118 | 15 | 15 | 18 | 10 | 4 |
| 11 | balance-scale | 625 | 499 | 63 | 63 | 4 | 10 | 3 |
| 12 | mfeat-factors | 2000 | 1600 | 200 | 200 | 216 | 10 | 10 |
| 14 | mfeat-fourier | 2000 | 1600 | 200 | 200 | 76 | 10 | 10 |
| 15 | breast-w | 699 | 559 | 70 | 70 | 9 | 10 | 2 |
| 16 | mfeat-karhunen | 2000 | 1600 | 200 | 200 | 64 | 10 | 10 |
| 18 | mfeat-morpholog... | 2000 | 1600 | 200 | 200 | 6 | 10 | 10 |
| 23 | cmc | 1473 | 1177 | 148 | 148 | 9 | 10 | 3 |
| 25 | colic | 368 | 294 | 37 | 37 | 26 | 10 | 2 |
| 27 | colic | 368 | 294 | 37 | 37 | 22 | 10 | 2 |
| 29 | credit-approval | 690 | 552 | 69 | 69 | 15 | 10 | 2 |
| 30 | page-blocks | 5473 | 4377 | 548 | 548 | 10 | 10 | 5 |
| 35 | dermatology | 366 | 292 | 37 | 37 | 34 | 10 | 6 |
| 37 | diabetes | 768 | 614 | 77 | 77 | 8 | 10 | 2 |
| 39 | sonar | 208 | 166 | 21 | 21 | 60 | 10 | 2 |
| 40 | glass | 214 | 170 | 22 | 22 | 9 | 10 | 6 |
| 43 | spambase | 4601 | 3680 | 460 | 461 | 57 | 10 | 2 |
| 45 | splice | 3190 | 2552 | 319 | 319 | 60 | 10 | 3 |
| 47 | tae | 151 | 120 | 15 | 16 | 5 | 10 | 3 |
| 48 | heart-c | 303 | 241 | 31 | 31 | 13 | 10 | 2 |
| 49 | tic-tac-toe | 958 | 766 | 96 | 96 | 9 | 10 | 2 |
| 50 | heart-h | 294 | 234 | 30 | 30 | 13 | 10 | 2 |
| 53 | vehicle | 846 | 676 | 85 | 85 | 18 | 10 | 4 |
| 59 | iris | 150 | 120 | 15 | 15 | 4 | 10 | 3 |
| 2074 | satimage | 6430 | 5144 | 643 | 643 | 36 | 10 | 6 |
| 2079 | eucalyptus | 736 | 588 | 74 | 74 | 19 | 10 | 5 |
| 2867 | anneal | 898 | 718 | 90 | 90 | 38 | 10 | 5 |
| 3485 | scene | 2407 | 1925 | 241 | 241 | 299 | 10 | 2 |
| 3512 | synthetic_contr... | 600 | 480 | 60 | 60 | 60 | 10 | 6 |
| 3540 | analcatdata_box... | 120 | 96 | 12 | 12 | 3 | 10 | 2 |
| 3543 | irish | 500 | 400 | 50 | 50 | 5 | 10 | 2 |
| 3549 | analcatdata_aut... | 841 | 672 | 84 | 85 | 70 | 10 | 4 |
| 3560 | analcatdata_dmf... | 797 | 637 | 80 | 80 | 4 | 10 | 6 |
| 3561 | profb | 672 | 536 | 68 | 68 | 9 | 10 | 2 |
| 3602 | visualizing_env... | 111 | 88 | 11 | 12 | 3 | 10 | 2 |
| 3620 | fri_c0_100_5 | 100 | 80 | 10 | 10 | 5 | 10 | 2 |
| 3647 | rabe_266 | 120 | 96 | 12 | 12 | 2 | 10 | 2 |
| 3711 | elevators | 16599 | 13279 | 1660 | 1660 | 18 | 10 | 2 |
| 3731 | visualizing_liv... | 130 | 104 | 13 | 13 | 2 | 10 | 2 |
| 3739 | analcatdata_chl... | 100 | 80 | 10 | 10 | 3 | 10 | 2 |
| 3748 | transplant | 131 | 104 | 13 | 14 | 3 | 10 | 2 |
| 3779 | fri_c3_100_5 | 100 | 80 | 10 | 10 | 5 | 10 | 2 |
| 3797 | socmob | 1156 | 924 | 116 | 116 | 5 | 10 | 2 |
| 3896 | ada_agnostic | 4562 | 3648 | 457 | 457 | 48 | 10 | 2 |
| 3902 | pc4 | 1458 | 1166 | 146 | 146 | 37 | 10 | 2 |
| 3903 | pc3 | 1563 | 1249 | 157 | 157 | 37 | 10 | 2 |
| 3904 | jm1 | 10885 | 8707 | 1089 | 1089 | 21 | 10 | 2 |
| 3913 | kc2 | 522 | 416 | 53 | 53 | 21 | 10 | 2 |
| 3917 | kc1 | 2109 | 1687 | 211 | 211 | 21 | 10 | 2 |
| 3918 | pc1 | 1109 | 887 | 111 | 111 | 21 | 10 | 2 |
| 3953 | adult-census | 32561 | 26048 | 3256 | 3257 | 14 | 10 | 2 |
| 9946 | wdbc | 569 | 455 | 57 | 57 | 30 | 10 | 2 |
| 9952 | phoneme | 5404 | 4322 | 541 | 541 | 5 | 10 | 2 |
| 9957 | qsar-biodeg | 1055 | 843 | 106 | 106 | 41 | 10 | 2 |
| 9960 | wall-robot-navi... | 5456 | 4364 | 546 | 546 | 24 | 10 | 4 |
| 9964 | semeion | 1593 | 1273 | 160 | 160 | 256 | 10 | 10 |
| 9971 | ilpd | 583 | 465 | 59 | 59 | 10 | 10 | 2 |
| 9978 | ozone-level-8hr | 2534 | 2026 | 254 | 254 | 72 | 10 | 2 |
| 9984 | fertility | 100 | 80 | 10 | 10 | 9 | 10 | 2 |
| 10089 | acute-inflammat... | 120 | 96 | 12 | 12 | 6 | 10 | 2 |
| 10093 | banknote-authen... | 1372 | 1096 | 138 | 138 | 4 | 10 | 2 |
| 10101 | blood-transfusi... | 748 | 598 | 75 | 75 | 4 | 10 | 2 |
| 14952 | PhishingWebsite... | 11055 | 8843 | 1106 | 1106 | 30 | 10 | 2 |
| 14954 | cylinder-bands | 540 | 432 | 54 | 54 | 37 | 10 | 2 |

| 14965 | bank-marketing | 45211 | 36168 | 4521 | 4522 | 16 | 10 | 2 |
|---|---|---|---|---|---|---|---|---|
| 14967 | cjs | 2796 | 2236 | 280 | 280 | 33 | 10 | 6 |
| 125920 | dresses-sales | 500 | 400 | 50 | 50 | 12 | 10 | 2 |
| 125921 | LED-display-dom... | 500 | 400 | 50 | 50 | 7 | 10 | 10 |
| 145793 | yeast | 1269 | 1015 | 127 | 127 | 8 | 10 | 4 |
| 145799 | breast-cancer | 286 | 228 | 29 | 29 | 9 | 10 | 2 |
| 145836 | blood-transfusi... | 748 | 598 | 75 | 75 | 4 | 10 | 2 |
| 145847 | hill-valley | 1212 | 968 | 122 | 122 | 100 | 10 | 2 |
| 145977 | ecoli | 336 | 268 | 34 | 34 | 7 | 10 | 8 |
| 145984 | ionosphere | 351 | 280 | 35 | 36 | 34 | 10 | 2 |
| 146024 | lung-cancer | 32 | 24 | 4 | 4 | 56 | 10 | 3 |
| 146063 | hayes-roth | 160 | 128 | 16 | 16 | 4 | 10 | 3 |
| 146065 | monks-problems-... | 601 | 480 | 60 | 61 | 6 | 10 | 2 |
| 146192 | car-evaluation | 1728 | 1382 | 173 | 173 | 21 | 10 | 4 |
| 146210 | postoperative-p... | 88 | 70 | 9 | 9 | 8 | 10 | 2 |
| 146607 | SpeedDating | 8378 | 6702 | 838 | 838 | 120 | 10 | 2 |
| 146800 | MiceProtein | 1080 | 864 | 108 | 108 | 77 | 10 | 8 |
| 146817 | steel-plates-fa... | 1941 | 1552 | 194 | 195 | 27 | 10 | 7 |
| 146818 | Australian | 690 | 552 | 69 | 69 | 14 | 10 | 2 |
| 146820 | wilt | 4839 | 3871 | 484 | 484 | 5 | 10 | 2 |
| 146821 | car | 1728 | 1382 | 173 | 173 | 6 | 10 | 4 |
| 167140 | dna | 3186 | 2548 | 319 | 319 | 180 | 10 | 3 |
| 167141 | churn | 5000 | 4000 | 500 | 500 | 20 | 10 | 2 |
| 167211 | Satellite | 5100 | 4080 | 510 | 510 | 36 | 10 | 2 |
| 168911 | jasmine | 2984 | 2386 | 299 | 299 | 144 | 10 | 2 |
| 190408 | Click_predictio... | 39948 | 31958 | 3995 | 3995 | 11 | 10 | 2 |
| 360948 | libras | 360 | 288 | 36 | 36 | 104 | 10 | 10 |

## A.4 RUN TIMES

Tables 6 and 7 present the run times for Attic, TabForestPFN and AutoGluon. The ICL-transformers are run on an H100, while AutoGluon is run on 64 cores of a Intel Xeon Gold 5220R CPU. Auto-Gluon is only run on the WhyTrees benchmark due to the high run times. We would like to empha-size that the run time of AutoGluon should only be compared to that of ICL-transformers by orders of magnitude. As AutoGluon runs on CPUs and the ICL-transformers run on GPUs, any direct run-time comparison critically depends on the equipment used.

Table 6: Run times of TabForestPFN, Attic and AutoGluon on the WhyTrees benchmark. The runtime is the end-to-end time in seconds for one cross validation split. End-to-end time includes loading, preprocessing, training and testing.

| Data | | Size | | TabForestPFN | | Attic | | AutoGluon |
|---|---|---|---|---|---|---|---|---|
| OpenML | | | | | | | | |
| ID | Name | Obs. | Feat. | Zero-shot | Fine-tuned | Zero-shot | Fine-tuned | Best-Quality |
| 44089 | credit | 10000 | 10 | 9 | 103 | 12 | 90 | 5982 |
| 44120 | electricity | 10000 | 7 | 15 | 151 | 18 | 125 | 10390 |
| 44121 | covertype | 10000 | 10 | 34 | 167 | 36 | 146 | 45419 |
| 44122 | pol | 7057 | 26 | 6 | 57 | 11 | 104 | 4714 |
| 44123 | house_16H | 9441 | 16 | 8 | 72 | 12 | 111 | 7081 |
| 44125 | MagicTelescope | 9363 | 10 | 7 | 105 | 11 | 103 | 7188 |
| 44126 | bank-marketing | 7404 | 7 | 7 | 68 | 9 | 49 | 5198 |
| 44128 | MiniBooNE | 10000 | 50 | 28 | 126 | 75 | 376 | 8119 |
| 44129 | Higgs | 10000 | 24 | 34 | 119 | 56 | 219 | 26781 |
| 44130 | eye_movements | 5325 | 20 | 5 | 63 | 9 | 140 | 6311 |
| 44156 | electricity | 10000 | 8 | 17 | 142 | 19 | 131 | 16432 |
| 44157 | eye_movements | 5325 | 23 | 6 | 65 | 10 | 149 | 4896 |
| 44159 | covertype | 10000 | 54 | 37 | 219 | 103 | 476 | 45452 |
| 45019 | Bioresponse | 2403 | 419 | 8 | 34 | 19 | 333 | 4858 |
| 45020 | default-of-credit-card-clients | 9290 | 20 | 7 | 81 | 13 | 119 | 4839 |
| 45021 | jannis | 10000 | 54 | 23 | 130 | 63 | 457 | 8992 |

| 45022 | Diabetes130US | 10000 | 7 | 25 | 95 | 24 | 87 | 8536 |
|---|---|---|---|---|---|---|---|---|
| 45026 | heloc | 7000 | 22 | 6 | 56 | 11 | 85 | 6490 |
| 45028 | california | 10000 | 8 | 11 | 112 | 15 | 82 | 7022 |
| 45035 | albert | 10000 | 31 | 21 | 103 | 40 | 252 | 8116 |
| 45036 | default-of-credit-card-clients | 9290 | 21 | 8 | 79 | 13 | 135 | 4648 |
| 45038 | road-safety | 10000 | 32 | 30 | 153 | 60 | 349 | 14055 |
| 45039 | compas-two-years | 3476 | 11 | 5 | 43 | 8 | 41 | 3579 |

Table 7: Run times of TabForestPFN and the Attic on the TabZilla benchmark. The runtime is the end-to-end time in seconds for one cross validation split. End-to-end time includes loading, preprocessing, training and testing.

| Data | | | | Run time (s) | | | |
|---|---|---|---|---|---|---|---|
| OpenML | | Size | | TabForestPFN | | Attic | |
| ID | Name | Obs. | Feat. | Zero-shot | Fine-tuned | Zero-shot | Fine-tuned |
| 3 | kr-vs-kp | 2556 | 36 | 4 | 29 | 8 | 53 |
| 4 | labor | 45 | 16 | 3 | 13 | 9 | 25 |
| 9 | autos | 163 | 25 | 3 | 11 | 8 | 18 |
| 10 | lymph | 118 | 18 | 3 | 9 | 8 | 18 |
| 11 | balance-scale | 499 | 4 | 3 | 32 | 7 | 28 |
| 12 | mfeat-factors | 1600 | 216 | 5 | 26 | 13 | 145 |
| 14 | mfeat-fourier | 1600 | 76 | 4 | 29 | 10 | 47 |
| 15 | breast-w | 559 | 9 | 3 | 19 | 8 | 19 |
| 16 | mfeat-karhunen | 1600 | 64 | 4 | 22 | 10 | 42 |
| 18 | mfeat-morphological | 1600 | 6 | 4 | 22 | 8 | 19 |
| 23 | cmc | 1177 | 9 | 4 | 20 | 8 | 18 |
| 25 | colic | 294 | 26 | 3 | 10 | 8 | 16 |
| 27 | colic | 294 | 22 | 3 | 11 | 8 | 18 |
| 29 | credit-approval | 552 | 15 | 4 | 22 | 7 | 17 |
| 30 | page-blocks | 4377 | 10 | 5 | 40 | 8 | 37 |
| 35 | dermatology | 292 | 34 | 3 | 13 | 7 | 22 |
| 37 | diabetes | 614 | 8 | 3 | 19 | 7 | 17 |
| 39 | sonar | 166 | 60 | 3 | 11 | 7 | 19 |
| 40 | glass | 170 | 9 | 3 | 10 | 7 | 17 |
| 43 | spambase | 3680 | 57 | 6 | 42 | 12 | 92 |
| 45 | splice | 2552 | 60 | 3 | 33 | 9 | 83 |
| 47 | tae | 120 | 5 | 3 | 11 | 8 | 22 |
| 48 | heart-c | 241 | 13 | 3 | 11 | 8 | 19 |
| 49 | tic-tac-toe | 766 | 9 | 3 | 20 | 8 | 20 |
| 50 | heart-h | 234 | 13 | 2 | 12 | 8 | 16 |
| 53 | vehicle | 676 | 18 | 3 | 23 | 8 | 18 |
| 59 | iris | 120 | 4 | 3 | 16 | 8 | 19 |
| 2074 | satimage | 5144 | 36 | 6 | 55 | 11 | 118 |
| 2079 | eucalyptus | 588 | 19 | 3 | 18 | 7 | 17 |
| 2867 | anneal | 718 | 38 | 3 | 26 | 7 | 26 |
| 3485 | scene | 1925 | 299 | 6 | 37 | 15 | 201 |
| 3512 | synthetic_control | 480 | 60 | 3 | 18 | 7 | 22 |
| 3540 | analcatdata_boxing1 | 96 | 3 | 3 | 12 | 8 | 25 |
| 3543 | irish | 400 | 5 | 4 | 19 | 8 | 16 |
| 3549 | analcatdata_authorship | 672 | 70 | 4 | 25 | 9 | 32 |
| 3560 | analcatdata_dmft | 637 | 4 | 3 | 20 | 8 | 18 |
| 3561 | profb | 536 | 9 | 3 | 16 | 7 | 17 |
| 3602 | visualizing_environmental | 88 | 3 | 3 | 10 | 8 | 19 |
| 3620 | fri_c0_100_5 | 80 | 5 | 3 | 13 | 7 | 22 |
| 3647 | rabe_266 | 96 | 2 | 3 | 14 | 8 | 22 |
| 3711 | elevators | 13279 | 18 | 9 | 101 | 15 | 175 |
| 3731 | visualizing_livestock | 104 | 2 | 3 | 15 | 9 | 21 |
| 3739 | analcatdata_chlamydia | 80 | 3 | 3 | 16 | 8 | 22 |
| 3748 | transplant | 104 | 3 | 3 | 12 | 7 | 22 |
| 3779 | fri_c3_100_5 | 80 | 5 | 3 | 13 | 7 | 25 |

| 3797 | socmob | 924 | 5 | 3 | 19 | 8 | 14 |
|---|---|---|---|---|---|---|---|
| 3896 | ada_agnostic | 3648 | 48 | 6 | 36 | 11 | 82 |
| 3902 | pc4 | 1166 | 37 | 4 | 23 | 8 | 30 |
| 3903 | pc3 | 1249 | 37 | 4 | 26 | 9 | 34 |
| 3904 | jm1 | 8707 | 21 | 6 | 117 | 10 | 161 |
| 3913 | kc2 | 416 | 21 | 3 | 26 | 7 | 18 |
| 3917 | kc1 | 1687 | 21 | 4 | 48 | 8 | 27 |
| 3918 | pc1 | 887 | 21 | 3 | 16 | 8 | 18 |
| 3953 | adult-census | 26048 | 14 | 14 | 175 | 19 | 166 |
| 9946 | wdbc | 455 | 30 | 4 | 17 | 8 | 19 |
| 9952 | phoneme | 4322 | 5 | 4 | 44 | 8 | 22 |
| 9957 | qsar-biodeg | 843 | 41 | 4 | 23 | 7 | 20 |
| 9960 | wall-robot-navigation | 4364 | 24 | 5 | 42 | 9 | 65 |
| 9964 | semeion | 1273 | 256 | 5 | 26 | 11 | 147 |
| 9971 | ilpd | 465 | 10 | 4 | 20 | 8 | 16 |
| 9978 | ozone-level-8hr | 2026 | 72 | 4 | 25 | 10 | 55 |
| 9984 | fertility | 80 | 9 | 3 | 13 | 8 | 18 |
| 10089 | acute-inflammations | 96 | 6 | 3 | 10 | 8 | 18 |
| 10093 | banknote-authentication | 1096 | 4 | 4 | 20 | 7 | 17 |
| 10101 | blood-transfusion-service-center | 598 | 4 | 3 | 20 | 9 | 16 |
| 14952 | PhishingWebsites | 8843 | 30 | 8 | 103 | 15 | 269 |
| 14954 | cylinder-bands | 432 | 37 | 4 | 16 | 7 | 18 |
| 14965 | bank-marketing | 36168 | 16 | 17 | 165 | 24 | 234 |
| 14967 | cjs | 2236 | 33 | 4 | 79 | 10 | 47 |
| 125920 | dresses-sales | 400 | 12 | 4 | 18 | 7 | 18 |
| 125921 | LED-display-domain-7digit | 400 | 7 | 4 | 16 | 9 | 18 |
| 145793 | yeast | 1015 | 8 | 4 | 19 | 8 | 19 |
| 145799 | breast-cancer | 228 | 9 | 3 | 11 | 7 | 22 |
| 145836 | blood-transfusion-service-center | 598 | 4 | 3 | 21 | 9 | 16 |
| 145847 | hill-valley | 968 | 100 | 4 | 47 | 7 | 69 |
| 145977 | ecoli | 268 | 7 | 3 | 12 | 8 | 16 |
| 145984 | ionosphere | 280 | 34 | 3 | 12 | 7 | 16 |
| 146024 | lung-cancer | 24 | 56 | 3 | 14 | 8 | 15 |
| 146063 | hayes-roth | 128 | 4 | 3 | 14 | 8 | 17 |
| 146065 | monks-problems-2 | 480 | 6 | 2 | 22 | 7 | 19 |
| 146192 | car-evaluation | 1382 | 21 | 4 | 27 | 9 | 33 |
| 146210 | postoperative-patient-data | 70 | 8 | 3 | 13 | 8 | 20 |
| 146607 | SpeedDating | 6702 | 120 | 6 | 57 | 23 | 456 |
| 146800 | MiceProtein | 864 | 77 | 4 | 28 | 8 | 38 |
| 146817 | steel-plates-fault | 1552 | 27 | 4 | 22 | 9 | 30 |
| 146818 | Australian | 552 | 14 | 4 | 23 | 8 | 16 |
| 146820 | wilt | 3871 | 5 | 4 | 30 | 9 | 20 |
| 146821 | car | 1382 | 6 | 4 | 30 | 9 | 33 |
| 167140 | dna | 2548 | 180 | 4 | 26 | 13 | 208 |
| 167141 | churn | 4000 | 20 | 5 | 41 | 10 | 47 |
| 167211 | Satellite | 4080 | 36 | 5 | 40 | 11 | 67 |
| 168911 | jasmine | 2386 | 144 | 4 | 36 | 12 | 132 |
| 190408 | Click_prediction_small | 31958 | 11 | 14 | 129 | 20 | 182 |
| 360948 | libras | 288 | 104 | 3 | 11 | 8 | 21 |

## A.5 WHYTREES INDIVIDUAL DATASETS

In Figure 2 we have seen the average normalized accuracy score for various methods on the WhyTrees benchmark. In Figures 7 and 8 we show the performance on each individual benchmark. For regression, the individual datasets are shown in Figures 9 and 10

## A.6 ATTIC ZERO-SHOT COMPARISON

In Section 4.7 we have seen the comparison between fine-tuned Attic and other methods. In Figure 11 we show the comparison between zero-shot Attic and XGBoost, CatBoost, zero-shot TabForestPFN, and fine-tuned Attic.

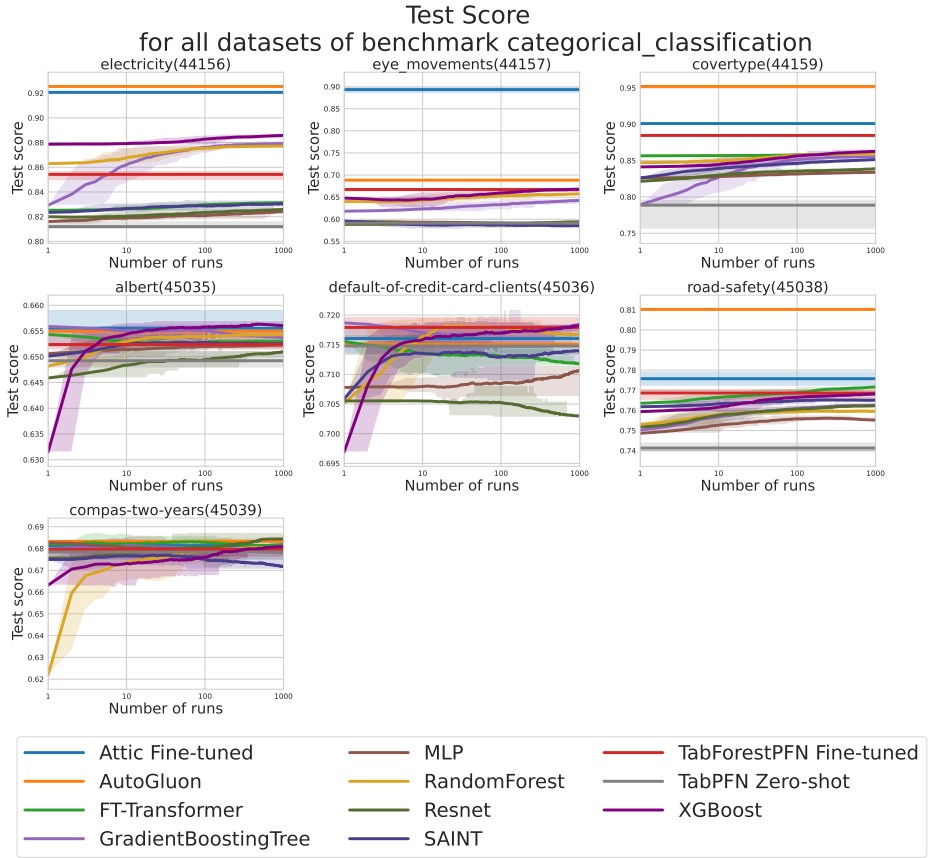

Figure 7: Performance of various classification methods on the WhyTrees benchmark with mixed features.

## A.7 COMPLEXITY SCORE

Breejen et al. (2024) use a complexity score $V$ to measure the complexity of decision boundaries. We produced the graphs and scores using their code, using the following formula for the complexity score:

$$V = \frac{1}{n} \sum_{ij} |p_{i+1,j} - p_{ij}| + |p_{i-1,j} - p_{i,j}| + |p_{i,j+1} - p_{ij}| + |p_{i,j-1} - p_{ij}|$$

Here $p_{i,j}$ is the prediction value of grid cell of feature space. As the analysis is done on two variables, the feature space is cut into grid cell, of which the middle point of the grid cell as an observation to predict. The complexity score then measures how fast the prediction changes when we move along the grid.

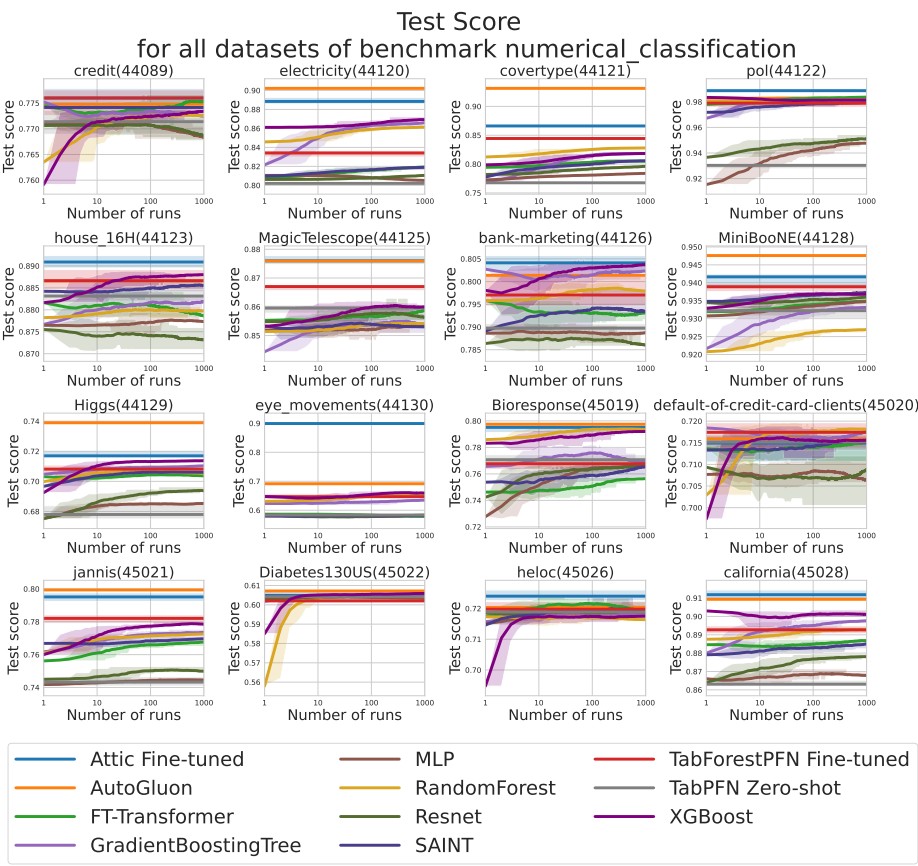

Figure 8: Performance of various classification methods on the WhyTrees benchmark with numerical features.

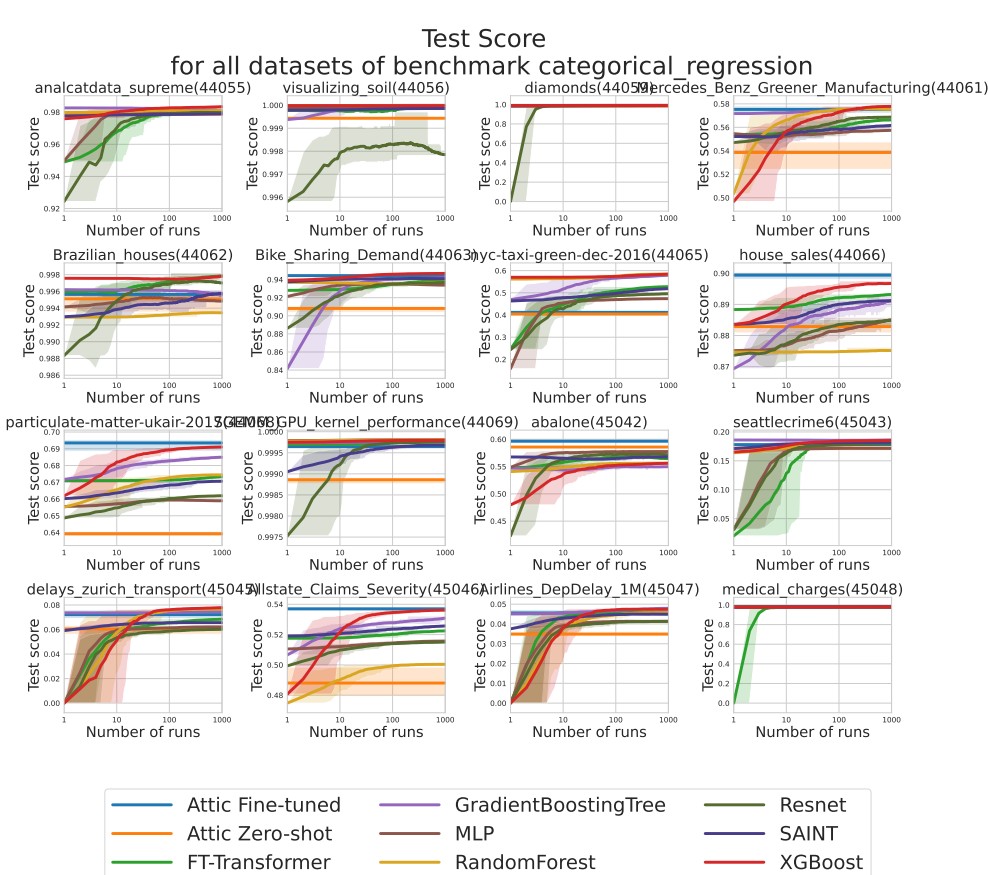

Figure 9: Performance of various regression methods on the WhyTrees benchmark with mixed features.

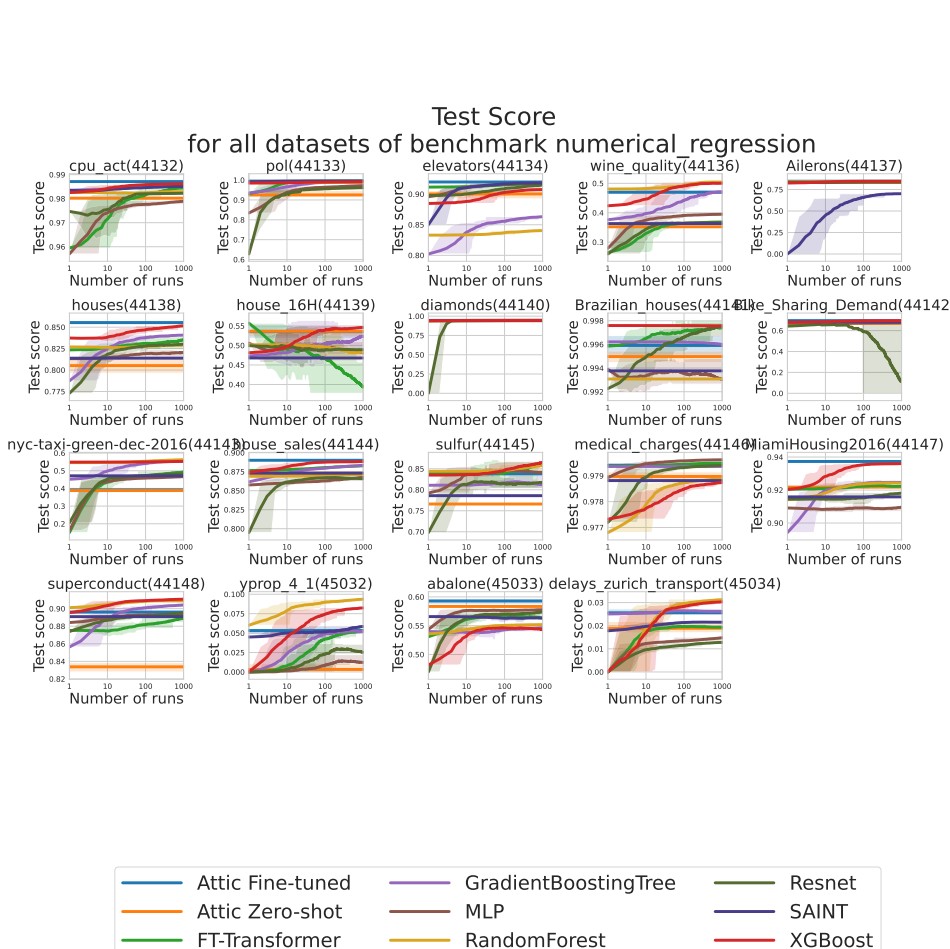

Figure 10: Performance of various regression methods on the WhyTrees benchmark with numerical features.

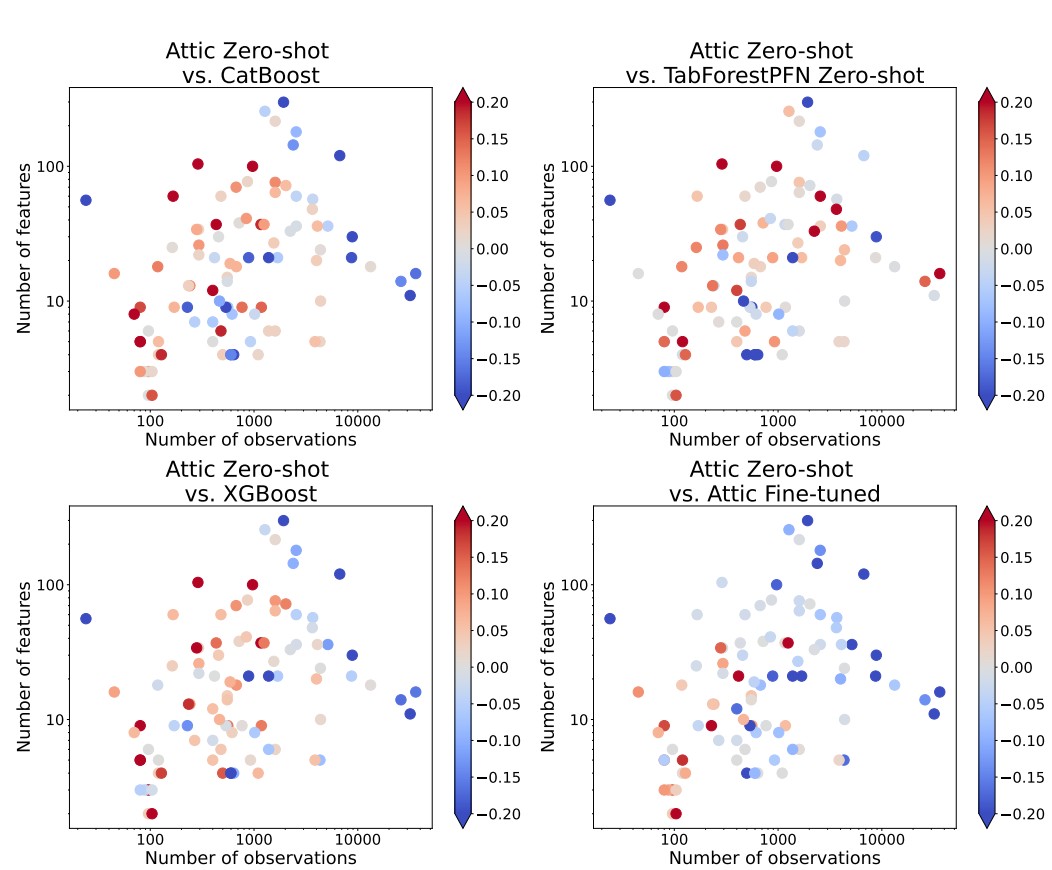

Figure 11: Comparison of fine-tuned Attic with XGBoost, CatBoost, TabForestPFN and it's zero-shot variant. Dots represent difference in normalized accuracy for an individual dataset from TabZilla. Red means Attic Zero-shot is better.

## A.8 TABZILLA CRITICAL DIFFERENCE DIAGRAMS

Here we present the critical difference diagrams for the TabZilla benchmark as shown in Table 2. We construct the diagrams following the procedure of the TabZilla authors McElfresh et al. (2023). The critical difference diagrams are constructed using paired Wilcoxon signed-rank tests under $\alpha = 0.05$ with Holm-Bonferroni correction to account for multiple comparisons. The tests are performed on the cross-entropy loss.

Using 94 datasets and 28 methods, the statistical power to compare all 28 methods is relatively low. Therefore, we created two versions of the critical difference diagram. Figure 12 considers all methods, while Figure 13 considers only the most important methods. The most important methods are selected as follows: we select the two most commonly used tree-based algorithms (XGBoost, CatBoost), the two most commonly used neural network based methods (MLP, FT-Transformer), Attic (Fine-tuned, Zero-Shot) and TabForestPFN (Fine-tuned, Zero-Shot).

In this setting, the two Attic versions are significantly different from the other six methods. In particular, the Wilcoxon signed-rank test rejects Attic Fine-tuned to have the same mean as TabForestPFN Fine-tuned with probability $P < 0.000000$ and rejects Attic Zero-shot to have the same mean as TabForestPFN Zero-shot with probability $P < 0.000006$.

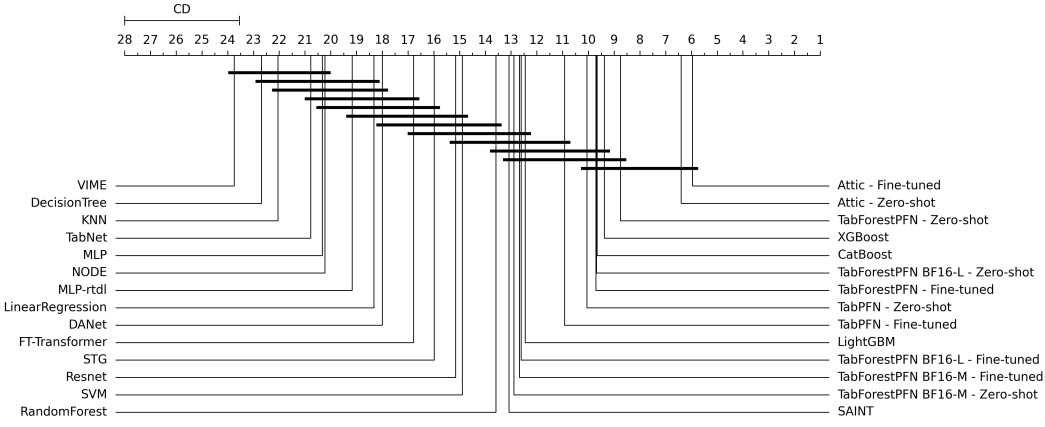

Figure 12: Critical difference diagram for 94 datasets of the TabZilla benchmark. All methods 28 included. Top row presents the relative rank of the methods ranked by the log loss. Bars show the significant bands: all methods within a band are not significantly different. Diagram made using the *autorank* package.

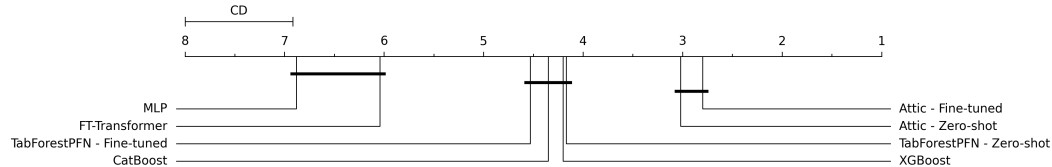

Figure 13: Critical difference diagram for 94 datasets of the TabZilla benchmark. Only the most important methods are included. Top row presents the relative rank of the methods ranked by the log loss. Bars show the significant bands: all methods within a band are not significantly different. Diagram made using the *autorank* package.

