# OpenReview forum: "Attic: A New Architecture for Tabular In-Context Learning Transformers"
_ICLR.cc/2025/Conference — Submitted to ICLR 2025_

### Official Review · Reviewer_FvnR · 2024-11-01

**Soundness:** 3
**Presentation:** 3
**Contribution:** 3
**Rating:** 5
**Confidence:** 3

**Summary:**

Traditional ICL transformers like TabPFN and TabForestPFN use observation tokens, which can be influenced by the order of these features. To overcome these challenges, this paper introduces Attic, which employs cell tokens to represent individual observations x features, allowing the model to handle features without being affected by their order. As a result, the authors claim that Attic significantly outperforms traditional methods like XGBoost, CatBoost, and TabPFN in benchmark tests from WhyTables and Tabzilla. These findings suggest that Attic is a strong candidate, given its superior accuracy on various datasets.

**Strengths:**

- The motivation is clear and well-founded. I like the idea of removing the feature order importance that is placed on classic ICL transformers. This is a nice thing to focus on and is a strength of this paper.
- I appreciate the author’s writing style and delivery of the material. The paper reads well and the authors provide a nice, logical flow for their arguments. (See below for some suggestions for additional details which will improve the paper even more!)
- The authors’ results are promising and they certainly show their motivation and technical solution (observation x feature tokens) is a promising research direction!
- Evaluating their method against existing benchmarks (Tabzilla and WhyTables) is a very important and necessary step for any new method in the tabular data space. I appreciate the authors' care and consideration here.

**Weaknesses:**

- More details are needed throughout the paper. For example, can the authors provide values instead of variables for architecture details:
    - Token dictionary size, what is the value of $L$, dimension sizes, what values of $k$ did you test?etc
    - How is the tokenization performed for each observation x feature?
    - What are the model sizes for the -M and -L versions of TabForestPFN?
- You claim in L153: “We changed this because we believe this formulation is more natural, but performance-wise, it has little impact.” Please provide an ablation describing and proving this claim.
- The authors need to highlight more clearly for the reader that this work and its findings are limited to datasets with 10 or fewer classes.
- Do the authors re-run TabPFN for the Tabzilla benchmark? or do they take the TabPFN results from the Tabzilla paper/results?
- What model is being reported in Tables 6 and 7 for TabForestPFN when you describe at least 3 different versions you use in your main paper? This actually is an important question/distinction the authors need to make through the paper. It is very hard to evaluate the paper without knowing this since the 3 versions of TabForestPFN are so different and the evaluation of the paper rests on understanding which version is being discussed in each results section. With further clarity, I’ll be able to update my score accordingly.
- Can the authors confirm the train/val/test splits they use are consistent with what the authors used in WhyTress and Tabzilla? My concern here is that tabular data use cases are not limited in the ways the authors may have selected their subset of Tabzilla. Tabzilla, which quickly has become a gold-standard benchmark in tabular data evaluations, is vast and captures a lot of nuance in the tabular community and use cases. I’m concerned the author’s down-selecting is increasing hiding some flaws of the approach.
- Can the authors please describe why they omit about half of the Tabzilla benchmark?
- Can the authors provide details on the computation resources? They provide comparisons to TabForestPFN primarily, but they should also provide comparisons to TabPFN and CatBoost at least — preferably more. Computational complexity is one of the main downsides of this approach, so the authors need to provide more care here for us to evaluate the work appropriately.

**Questions:**

See Weaknesses. I am certain that with satisfying clarifications on the questions above, I will happily raise my score.

Also, a minor typos:
- L221: “Additionally, pretraining Attic on smaller datasets than TabForestPFN favors TabForestPFN”. This typo makes the entirely of the paragraph of L219 hard to comprehend.

---

> ### Author Response · Authors · 2024-11-19
> **Rebuttal - Part 1**
>
> Dear Reviewer,
>
> Thank you for your review. In general, we followed the experimental setup of TabForestPFN and only reported differences in our setup, which might have lead to a lack of details in some parts of our paper. We would like to clarify any remaining points here.
>
> > Can the authors provide values instead of variables for architecture details: Token dictionary size L, what is the value of k, dimension sizes, what values of did you test?
> >
>
> All architectural details are presented in Table 1. The model has 12 layers, each with a dimension of 512. We kept the same dimensions as Tab(Forest)PFN and did not test other values.
>
> > How is the tokenization performed for each observation x feature?
> >
>
> Each observation × feature passes through the same linear layer, which has dimension 1x512.
>
> > What are the model sizes for the -M and -L versions of TabForestPFN?
> >
>
> These details are also presented in Table 1. The -M version has 12 layers with a dimension of 512, and the -L version has 24 layers with a dimension of 2048.
>
> > You claim in L153: “We changed this because we believe this formulation is more natural, but performance-wise, it has little impact.” Please provide an ablation describing and proving this claim.
> >
>
> When digging through the embedding of TabPFN, we found two unfamiliar design choices. First of all, TabPFN embeds the target label $y \in \cal{N}$ with a linear layer, as if it were numerical. Secondly, TabPFN uses feature count scaling: it multiplies all $x$ values with $\frac{100}{k}$, where $k$ is the number of features in $x$. In contrast, Attic does not feature count scale and embeds $y$ with a natural embedding layer for each class as is done in large language models.
>
> As requested, below is the ablation. The table corresponds to Table 3 in our paper. We change the embedding of TabForestPFN to the Attic embedding without feature count scaling. Due to the inclusion of the new model, the normalized accuracies for the original TabForestPFN and Attic are slightly different from Table 3. All in all, we consider the change of embedding insignificant.
>
> |  | Whytrees Mixed | WhyTrees Numerical |
> | --- | --- | --- |
> | **Zero-Shot** | - | - |
> | TabForestPFN (original embedding) | 0.416 | 0.597 |
> | TabForestPFN (Attic embedding) | 0.428 | 0.592 |
> | Attic | **0.444** | **0.610** |
> | **Fine-Tuned** | - | - |
> | TabForestPFN (original embedding) | 0.651       | 0.734 |
> | TabForestPFN (Attic embedding) | 0.641       | 0.754 |
> | Attic | **0.829**       | **0.890** |
>
> > The authors need to highlight more clearly for the reader that this work and its findings are limited to datasets with 10 or fewer classes.
> >
>
> We will explicitly add this information to Table 1.
>
> > Do the authors re-run TabPFN for the Tabzilla benchmark? or do they take the TabPFN results from the Tabzilla paper/results?
> >
>
> We re-ran TabPFN for the TabZilla benchmark. The TabZilla authors ran TabPFN on only 57 out of 94 datasets (those with fewer than 1,250 observations). In contrast, we ran TabPFN on all 94 datasets, including those with more than 1,250 observations.

---

> > ### Author Response · Authors · 2024-11-19
> > **Rebuttal - Part 2**
> >
> > > What model is being reported in Tables 6 and 7 for TabForestPFN when you describe at least 3 different versions you use in your main paper?
> > >
> >
> > We describe three versions of TabForestPFN: “TabForestPFN,” “TabForestPFN BF16 M,” and “TabForestPFN BF16 L.” Whenever “BF16 M” and “BF16 L” are not explicitly mentioned, the results refer to the original “TabForestPFN.” Tables 6 and 7, therefore, refer to the original TabForestPFN. Only Tables 2 and 3 include results for the BF16 M/L versions.
> >
> > The BF16 M version was excluded from further analysis due to extremely poor performance, while the BF16 L version was excluded because it showed worse performance and slower pretraining/inference times compared to the original. Therefore, we report results only for the original TabForestPFN to focus on the best-performing version
> >
> > > Can the authors confirm the train/val/test splits they use are consistent with what the authors used in WhyTrees and Tabzilla?
> > >
> >
> > We confirm that the splits are consistent. We generated the datasets using the original code from the WhyTrees and TabZilla authors and saved the train/validation/test splits as generated, which in case of TabZilla, includes ten different cross-validation splits. These exact splits where used throughout our whole research process. We also tested several algorithms to confirm the accuracies matched the public results.
> >
> > > Can the authors please describe why they omit about half of the Tabzilla benchmark?
> > >
> >
> > The short answer is that the TabZilla authors also omitted about half of their benchmark. In their paper, they used 98 out of 176 datasets because some datasets were too large for certain algorithms to complete training. All tables and figures in the main body of the original TabZilla paper are based on these 98 datasets. We used 94 datasets because 4 of the 98 datasets involved multiclass classification with more than 10 classes, which ICL-transformers do not currently support.
> >
> > > Can the authors provide details on the computation resources? They provide comparisons to TabForestPFN primarily, but they should also provide comparisons to TabPFN and CatBoost at least — preferably more.
> > >
> >
> > TabForestPFN and TabPFN differ only in their pretraining datasets, so they have identical runtime and computational requirements. For CatBoost and XGBoost, a fair comparison to ICL-transformers is difficult because CatBoost and XGBoost use CPUs, while ICL-transformers use GPUs. Hardware differences could lead to discrepancies of over one or two orders of magnitude, depending on the hardware model and the number of used cores/gpus, making a direct comparison unreliable. We decided not to provide comparisons that are so highly hardware-dependent.
> >
> > Also, further runtime reductions could be achieved using standard AI techniques such as quantization, pruning, distillation to smaller networks, or caching the query-context on the GPU to avoid redundant computation. We consider these techniques to be out of scope for our paper; however, applying these technique would also reduce the run time and affect the comparison between Attic and CPU methods.
> >
> > We will incorporate all extra information in the final version of the paper and the appendix. If there are any additional questions or clarifications required, please do not hesitate to ask.

---

> > > ### Comment · Reviewer_FvnR · 2024-11-26
> > > **Thank you**
> > >
> > > I'd like to thank the authors for their response to my and the other reviewer's questions and concerns. A few comments:
> > > - I wish the authors had updated the paper with the work that they promised. I particularly was hoping to see the statistical analyses and reproduction of TabZilla Figure 2; but I was discouraged that the authors didn't show their work in this assertion.
> > > - All the reviewers thought the lack of runtime / complexity analysis and comparison was a significant weakness. I totally understand and agree with the author's point that the purpose of new research isn't to make an algorithm the most efficient it can be. That barrier is too high and I don’t think is what I and the reviewers are looking for. I think it is entirely a reasonable expectations for authors who propose a new method that is more computationally complex to provide more details on this in the main paper.
> > >
> > > I have raised my score and hope the authors put serious effort into re-writing major portions of the paper to more effectively communicate their ideas in line with the concerns and confusion of the reviewers.

---

> > > > ### Author Response · Authors · 2024-11-26
> > > > **Thank you**
> > > >
> > > > Dear Reviewer,
> > > >
> > > > Thank you for your response.
> > > >
> > > > We were waiting for the responses from all reviewers before updating the paper. We thought it would be best to first confirm that we understood all the reviews correctly and then update the paper accordingly. Unfortunately, since the responses came at the last possible moment, we were unable to update the majority of the writing before the editing deadline. Rest assured, all sections of the paper will be updated to ensure any confusion or missing information is addressed.
> > > >
> > > > Regarding the statistical analysis of Table 2 (TabZilla results), reviewer 56pB asked for the significance tests between Attic and TabForestPFN. We addressed the key points in the rebuttal text. However, we were unaware that you, as a reviewer, also wanted to see more details of the statistical analysis. We have just uploaded a new version of the paper; please refer to the very last page (Appendix A.8) for the full statistical analysis. Other than the last page, no significant changes have been made. With the extended deadline, there is still time to discuss this new analysis in case it raises any questions.

---

### Official Review · Reviewer_56pB · 2024-11-01

**Soundness:** 2
**Presentation:** 3
**Contribution:** 3
**Rating:** 6
**Confidence:** 4

**Summary:**

The paper is on in context learning (ICL) for tabular prediction problems.
Essentially the complete tabular training data is placed in context and predictions are performed on test data. The main difference to existing work is that in the proposed approach each cell value is represented individually as compared to an entire row.
The approach is evaluated on both classification and regression problems and compared to existing ICL approaches as well as standard ML models such as gradient boosted trees.

**Strengths:**

In theory the idea of representing feature values individually seems very reasonable to me since it is allows a richer representation of the data. They also show that extending the existing architectures (TabPFN/TabForestPFN) accordingly can be performed pretty smoothly (Figure 1).

The evaluation is performed on more than 100 benchmarks and the authors consider both classification and regression problems.

The performance on classification problems (see Table 2 and Table3) seems to outperform a wide range of existing approaches.

Section 4.6 and 4.7 provide interesting insights into the decision boundaries obtained by the proposed models as well as some indication on which kind of benchmarks the approach does not work so well.

**Weaknesses:**

First, to me there are the following limitations of the proposed approach:
1. Representing each value is sensible, but it does come at a substantial computational cost. The growth is in the number of columns and there are data sets with more than a thousand columns. Now, one can argue that for the ICL setting there will always be limitations (e.g., 1 billion rows and 1k columns is unlikely to happen). However, when looking at Table 2 TabForestPFN is only marginally worse than Attic (ZeroShot) but much more cost effective due to the more abstract representation.
2. On regression tasks (Figure 6) the performance is by far not as good. And that is actually interesting and not necessarily something negative about this approach. While in theory classification and regression are the same, it does seem to pose issues especially for DL approaches in general and not just the approach presented here.


Moreover, there are some points in the paper that are unclear to me.
For instance, in Section 4.2 it states that 94 out of 176 benchmarks are used from TabZilla.
One question is how those were selected, but to me the more important question is how they are split up into classification and regression tasks (this is explained for the WhyTrees benchmarks but not TabZilla).
Looking at Table 2, it seems that there only 28 classification tasks.This number of benchmarks weakens the results in my opinion - especially given that on regression it is known not to work as well.

In terms of hyper-parameter optimization - what was the experimental setup? For instance, was there a split of train/validation to perform  HPO and then the evaluation on test? To me this is not stated clearly in the paper.
One could also pick a bit more sophisticated HPO approach (e.g., https://github.com/hyperopt/hyperopt-sklearn ), but at least random search is used (and not grid search).

At the end of Section 3 it is stated: "We believe that this dependency on feature order leads to training inefficiencies. The cell-token
ICL-transformer treats each feature the same and learns how to construct relationships between features."
And while I agree with this intuition, why not perform an ablation study on it (e.g., shuffling columns and see if it impacts the performance)?

Similarly, it would be useful to try to explain the results in Table 3 a bit more - is it that the model needs the fine-tuning to embed the categorical values appropriately? The gap between fine-tuned and zero-shot is very large and it would be great to understand the reason for it better.

**Questions:**

In addition to the above questions:

Is it possible to show if the differences in performances shown in Table 2 are statistically significant?

Figure 4 is interesting - but isn't it the case that as the decision boundaries become more and more detailed, the generalization capability of the approach is reduced at the same time?

In terms of computational complexity is there a way to show computational cost (e.g., as a new column in Table 2)?
The question here is of course what the metric in this context could be - maybe the combination of runtime/memory usage/GPU requirement (for GPU use: number of input/output tokens) would already be insightful.

---

> ### Author Response · Authors · 2024-11-19
> **Rebuttal**
>
> Dear Reviewer,
>
> Thank you for your review. We agree with the strengths you presented and will address the questions raised below.
>
> &nbsp;
>
> ### Weaknesses
>
> > However, when looking at Table 2 TabForestPFN is only marginally worse than Attic (ZeroShot) but much more cost effective due to the more abstract representation.
> >
>
> We do not believe the performance difference is marginal. The median rank improves from 10.0 to 6.2, and the average accuracy improves from 80.9% to 83.4% across 94 datasets.  This difference is significant; see the answer to the question about significance scores.
>
> > Moreover, there are some points in the paper that are unclear to me. (Regarding TabZilla benchmark)
> >
>
> The TabZilla dataset is a benchmark consisting of 176 classification datasets; there are no regression datasets included. However, the authors of TabZilla use only 98 datasets because many algorithms take too long to run on larger datasets, leaving incomplete results. Therefore, although TabZilla presents 176 datasets, it is more accurate to say the benchmark effectively includes 98 datasets. Since all ICL-transformers are limited to multiclass classification with a maximum of 10 classes, we remove 4 additional datasets, resulting in 94 datasets in total. The number 28 in the table refers to the number of methods tested, not the number of datasets.
>
> > In terms of hyper-parameter optimization - what was the experimental setup?
> >
>
> We followed the experimental setups described in the benchmark papers (TabZilla and WhyTrees). For non-ICL transformers, training was performed on the training dataset, with early-stopping/validation on the validation dataset. The reported test accuracy corresponds to the test accuracy of the best validation run. For ICL-transformers, early-stopping for fine-tuning was performed on the validation set without hyperparamer search.
>
> > At the end of Section 3 it is stated: "We believe that this dependency on feature order leads to training inefficiencies. The cell-token ICL-transformer treats each feature the same and learns how to construct relationships between features." And while I agree with this intuition, why not perform an ablation study on it (e.g., shuffling columns and see if it impacts the performance)?
> >
>
> We do not believe an ablation study can effectively demonstrate this inefficiency. For inference, shuffling the features does indeed lead to different predictions for feature-variant models. However, the focus of our argument in section 3 is about pretraining. Both Tab(Forest)PFN and Attic are pretrained on synthetic data generators. Shuffling the features during pretraining would make no difference because the synthetic data generator already outputs an infinite stream of datasets with randomly ordered features. To show that the pretraining of Tab(Forest)PFN is inefficient due to the feature-variance, one approach is to switch the feature-variant module for a feature-invariant module and compare the performance. This approach effectively boils down to creating Attic.
>
> &nbsp;
>
> ### Questions
>
> > Is it possible to show if the differences in performances shown in Table 2 are statistically significant?
> >
>
> We will add the critical difference diagram as done by Tabzilla to the appendix. Just like the Tabzilla authors, we use multiple Wilcoxon signed-rank tests with a Holm-Bonferroni correction on the log loss values. As a result, Finetuned Attic is significantly different from Finetuned TabForestPFN and Zero-Shot Attic is significantly different from Zero-Shot TabForestPFN.
>
> > Figure 4 is interesting - but isn't it the case that as the decision boundaries become more and more detailed, the generalization capability of the approach is reduced at the same time?
> >
>
> This is a great question, the answer is: not necessarily. If the decision boundaries of the model are more detailed than the “ground truth” decision boundaries, it indicates overfitting and harms the generalization performance. However, in Figure 4, the “ground truth” decision boundaries are inherently complex, so a good model should match that complexity. Figure 4 demonstrates that fine-tuned Attic is capable of creating complex decision boundaries when a specific dataset requires them, which in turn leads to better generalization (higher test accuracy). We want to show this figure because standard neural networks seem to struggle to create these complex decision boundaries.
>
> > In terms of computational complexity is there a way to show computational cost (e.g., as a new column in Table 2)?
> >
>
> We report run times in Appendix A.4, which we consider the appropriate metric for computational cost. Using FLOPS would be unfair, as some models use FlashAttention while others do not. Similarly, memory costs and token counts are incomplete metrics because datasets that do not fit into memory can still be processed using a limited context size.
>
> &nbsp;
>
> If there is anything else that remains unclear, please do not hesitate to ask further.

---

> > ### Comment · Reviewer_56pB · 2024-11-25
> >
> > Thank you for your reply.
> >
> > Yes, the improvements are not that 'marginal' on its own. I actually meant it mostly in the context of the additional computational cost relatively to TabForestPFN. And maybe pulling a small summary/discussion of this cost information from the appendix into the main paper might be useful.
> >
> > For the hyper-parameter optimization, please do make it more explicit in the paper. One would assume that HPO is performed on train/validation and then the HP settings are applied on test data evaluations, but it is not unusual for mistakes being made there. And sometimes related issues such as data leakage are not that obvious.
> >
> > Dependency on feature order: probably updating the sentences in the paper using your explanation from the rebuttal might help.
> >
> > Tabzilla benchmark: For future work the discussion in the following paper might also be a useful reference in terms of what datasets to choose etc. https://arxiv.org/pdf/2406.19380

---

> > > ### Author Response · Authors · 2024-11-26
> > > **Thank you**
> > >
> > > Dear Reviewer,
> > >
> > > Thank you for your response. We will make adjustments to the text regarding the computational cost, HPO and feature order. The data leakage issue is interesting, thank you for sharing this paper. It seems that this affects the commonly used tabular benchmarks; we will keep an eye out for future tabular benchmarks published.

---

### Official Review · Reviewer_QP5E · 2024-11-01

**Soundness:** 2
**Presentation:** 3
**Contribution:** 3
**Rating:** 6
**Confidence:** 3

**Summary:**

The authors propose a new tabular in-context learning algorithm. They introduce a new dimension in the model input by modeling each feature as one token rather than each sample. This allows for feature order invariance at the cost of extra computation. The proposed model, ATTIC, makes substantial performance gains.

**Strengths:**

- The experimental results of the method is really exciting. As someone very familiar with the baselines, ATTIC seems to address a major performance bottleneck in tabular in-context learning.

- The authors provide an simple intuitive approach, encode each feature seperately to overcome feature order invariance instead of using ensembling, which provides large performance improvements.

- The authors provide results on several hard benchmarks and achieve performance with no hyperparameter tuning. The decision boundaries results provide further evidence for ATTIC's efficacy.

- The authors provide a detailed comparison of Attic compared to TabForestPFN for easy reproducibility.

**Weaknesses:**

Overall, I think some more in-depth analysis into the runtime and memory tradeoffs Attic makes to achieve its superior performance can greatly benefit the paper.

- Because Attic introduces a new dimension over the feature counts, I imagine the runtime costs greatly increase. How much does this increase with respect to the number of features?

- Similar to the previous question. How does the memory costs of Attic rise with respect tothe number of features?

- My understanding is TabPFN overcomes feature order invariance through ensembling across multiple feature shufflings. Could you discuss the tradeoff between using larger TabPFN ensembles and Attic, as both incur additional runtime costs?

- Intuitively, I feel larger feature count datasets would benefit the most from Attic. Empirically, are there trends on what specific datasets Attic most improves upon baseline algorithms?

**Questions:**

See Weaknesses.

Will you open-source your code?

---

> ### Author Response · Authors · 2024-11-19
> **Rebuttal**
>
> Dear Reviewer,
>
> Thank you for your review. To address your last question, yes, all the code will be open-source. This includes everything: preprocessing scripts, exact training settings, loss curves, and figure-drawing scripts.
>
> &nbsp;
>
> ### Weaknesses
>
> > Overall, I think some more in-depth analysis into the runtime and memory tradeoffs Attic makes to achieve its superior performance can greatly benefit the paper.
> >
>
> While we do agree that efficiency is an interesting and relevant direction, we believe that high focus on maximum attainable accuracy is more important. In the development of large language models, the first objective was to make a model that achieves high quality, and after researchers were sufficiently pleased with the performance, the field largely shifted to efficiency.
>
> We believe we should approach tabular neural networks in the same way. First focus on achieving maximum accuracy, and after we as a field are satisfied with the accuracy, we can use classical tools like quantization, pruning, distillation and efficient attention mechanisms to improve the fine-tuning and inference run time metrics.
>
> > Because Attic introduces a new dimension over the feature counts, I imagine the runtime costs greatly increase. How much does this increase with respect to the number of features?
> >
>
> For $n$ observations, $k$ features and model dimension $d$, the computation complexity of Attic is given by $O(n^2kd + nk^2d + nkd^2)$ due to the observation attention, feature attention and linear layers respectively. As often $n > k$, we expect the term $n^2kd$ to dominate. In contrast, the computation TabForestPFN scales $O(n^2d + nd^2)$. However, TabForestPFN is limited to a 100 features. If we want to pretrain TabForestPFN for 200 features for example, we likely need more model dimension space $d$ to encompass all features, so the complexity still indirectly depends on the number of features.
>
> In Appendix A.4, we show the runtime for every single dataset in the benchmark for both Attic and TabForestPFN to provide a clearer picture of how the number of features affects runtime.
>
> > Similar to the previous question. How does the memory costs of Attic rise with respect to the number of features?
> >
>
> For the TabForestPFN, the attention mechanism has a memory footprint of  $O(n^2d)$ and the activations have complexity $O(nd)$. As Attic switches to FlashAttention, which has linear memory cost scaling, the complexity of the activations and attention mechanisms are both $O(nkd)$. As often $n > k$, we argue the memory costs scaling of Attic is not inferior to that of TabForestPFN.
>
> > My understanding is TabPFN overcomes feature order invariance through ensembling across multiple feature shufflings. Could you discuss the tradeoff between using larger TabPFN ensembles and Attic, as both incur additional runtime costs?
> >
>
> Ensembling is a technique used in the zero-shot inference of TabPFN. Because the zero-shot inference of Tab(Forest)PFN is lightweight, ensembling is indeed a cost-effective way to mitigate feature order variance. However, this ensembling technique is less advantageous during fine-tuning. During fine-tuning, we keep the feature order fixed, so after fine-tuning, the model can only use the feature order it was trained with. Ensembling cannot be applied without re-fine-tuning the model for each ensemble member. Since fine-tuned Attic is, on average, 2× slower than fine-tuned TabPFN, creating large ensembles of fine-tuned TabPFN would be significantly slower than using Attic.
>
> > Intuitively, I feel larger feature count datasets would benefit the most from Attic. Empirically, are there trends on what specific datasets Attic most improves upon baseline algorithms?
> >
>
> We agree with this intuition, but the empirical trend is slightly different. Based on Figure 5, which plots the TabZilla benchmark datasets, the number of observations is actually more important than the number of features. Fine-tuned Attic outperforms fine-tuned TabForestPFN on all datasets with more than a thousand observations, even those with relatively low feature counts.
>
> &nbsp;
>
> If there are any more questions, please don’t hesitate to ask.

---

> ### Comment · Reviewer_QP5E · 2024-11-26
> **Thank you for your response.**
>
> Thank you for your rebuttal! I maintain that the rough runtime is an important metric to report. Even if it is not good, you could mention it in the Limitations section, along with TabPFN's feature count limitations. I maintain my score based on my original review and ATTIC's impressive performance results.

---

> > ### Author Response · Authors · 2024-11-26
> > **Thank you**
> >
> > Dear Reviewer,
> >
> > Thank you for your response. We will add an extra paragraph with limitations to the conclusion section to highlight run time, memory consumption and feature count limitations.

---

### Official Review · Reviewer_m8oc · 2024-11-05

**Soundness:** 3
**Presentation:** 3
**Contribution:** 3
**Rating:** 6
**Confidence:** 4

**Summary:**

This paper presents Attic, a novel transformer architecture for tabular in-context learning that uses cell tokens instead of observation tokens. The work demonstrates significant empirical improvements over existing methods and makes a meaningful contribution to the field.

**Strengths:**

* The architectural modification from observation tokens to cell tokens is simple yet effective, showing substantial performance gains across multiple benchmarks
* Thorough empirical evaluation on two major benchmarks (WhyTrees and TabZilla) with comprehensive comparisons against state-of-the-art methods
* Technical contribution backed by clear motivation and intuitive explanations for why cell tokens may work better than observation tokens
* Impressive results showing Attic outperforming tree-based methods and ensemble approaches like AutoGluon on several datasets
* Detailed ablation studies and analysis of computational requirements and decision boundaries

**Weaknesses:**

1. Limited theoretical analysis or formal justification for why cell tokens perform better.
2. The authors note that Attic is significantly slower and more memory-intensive than TabForestPFN for datasets with many features. This scalability issue could limit Attic's practical applicability to large, high-dimensional datasets.
3. Mixed precision training issues are not fully resolved, with float16 training instability remaining unexplained.
4. Initial regression results feel preliminary and could be expanded.
5. More detailed analysis of failure cases where tree-based methods outperform Attic. Why, in datasets with less than 500 observations, does Attic underperform?
6. The paper does not thoroughly explore potential limitations or failure cases of Attic, which would provide a more balanced view of the method's capabilities.

**Questions:**

1. Any potential approaches to reduce memory and computational requirements of Attic?
2. How does Attic handle missing data or categorical variables? Are there any special preprocessing steps required?
3. Can you elaborate on Attic's performance in regression tasks compared to classification?

---

> ### Author Response · Authors · 2024-11-19
> **Rebuttal**
>
> Dear Reviewer, thank you for you review. We generally agree with the strengths and weaknesses outlined by the reviewer. Below, we address a few specific weaknesses and respond to the questions raised.
>
> &nbsp;
>
> ### Weaknesses
>
> > 2. The authors note that Attic is significantly slower and more memory-intensive than TabForestPFN for datasets with many features. This scalability issue could limit Attic's practical applicability to large, high-dimensional datasets.
> >
>
> It is true that Attic’s scalability is limited with respect to high-dimensional datasets. However, we would like to point out that TabPFN and TabForestPFN are also significantly limited in handling high-dimensional datasets due to their acceptance of a fixed maximum number of features (maximum of 100). If we want to pretrain Tab(Forest)PFN from scratch to be able to use a larger number of features, we likely need a larger model dimension to properly represent a larger number of features. Therefore, Tab(Forest)PFN has a similar scalability issue.
>
> > 5. More detailed analysis of failure cases where tree-based methods outperform Attic. Why, in datasets with less than 500 observations, does Attic underperform?
> 6. The paper does not thoroughly explore potential limitations or failure cases of Attic, which would provide a more balanced view of the method's capabilities.
> >
>
> While analyzing the failure cases of Attic would indeed be very interesting, it is also a challenging task. In Figures 5, 7 and 8 we can identify on which datasets Attic fails, but it is hard to analyse why it fails. With language and vision models, the output can be visualized or read for qualitative analysis, but with tabular data, it is hard to do any meaningful qualitative analysis due to the high dimensionality of the feature space. Many techniques like PCA or other statistical techniques often give limited insightful information, due to their dependence on linearity or other assumptions. If the reviewer has suggestions for suitable approaches or knows of relevant literature from tabular neural network papers that provide such analyses, we would greatly appreciate the guidance.
>
> &nbsp;
>
> ### Questions
>
> > 1. Any potential approaches to reduce memory and computational requirements of Attic?
> >
>
> We believe this will be an active area of future research. The answer also depends on which stage of the pipeline we aim to optimize for memory and computational efficiency. Pretraining will inherently have a high computational cost, but fine-tuning and inference could be improved using classic techniques such as quantization, pruning, distillation into a smaller network, or using more efficient attention mechanism. Additionally, caching the query-set observations for successive inferences could help reduce runtime.
>
> > 2. How does Attic handle missing data or categorical variables? Are there any special preprocessing steps required?
> >
>
> Currently, there is no special handling of missing values implemented. If preprocessing encounters a missing value, it fills the value with the column mean. Since the benchmark datasets used in this study do not contain missing values, this approach does not affect the reported performance metrics. However, handling missing values could be implemented relatively straightforwardly by pretraining Attic with a special missing-value mask. Due to the lack of established tabular missing-value benchmarks, we leave this direction open for future work.
>
> Categorical variables are not subject to any special preprocessing steps. Attic treats categorical variables as numerical variables, without applying one-hot encoding.
>
> > 3. Can you elaborate on Attic's performance in regression tasks compared to classification?
> >
>
> We observed that pretraining is less stable during regression tasks compared to classification tasks. While pretraining under the cross-entropy loss behaves similarly to training large language models, pretraining under the mean-squared error loss occasionally exhibits extreme outliers, where the loss sometimes spikes to values is high as 10^8 and crashes regularly. We are uncertain why the mean-squared error loss behaves differently from the cross-entropy loss, but we included these regression results to remain transparent about the issue.
>
> &nbsp;
>
> If there are any more questions, please don’t hesitate to ask.

---

> > ### Comment · Reviewer_m8oc · 2024-11-28
> >
> > Thank you for your detailed response. Overall, your responses have addressed many of my concerns, though I believe further investigation of failure cases would strengthen the work.

---

### Author Response · Authors · 2024-12-04
**Final Statement**

Dear Reviewers,

As the rebuttal period comes to a close, we would like to express our gratitude for the valuable reviews and feedback we received.

Our paper introduces a new architecture for tabular in-context learning, with the primary modification being a switch from an observation-token representation to a cell-token representation. This change resulted in a significant performance increase under the same pretraining compute budget. Based on the reviews and our subsequent discussions during the rebuttal, we would like to summarize the strengths and weaknesses highlighted by multiple reviewers.

**Strengths**

All reviewers agreed on the strengths of the paper. They praised the clear motivation of the method as well as the logical flow of the work. The token modification was described as intuitive and interesting. Additionally, the reviewers were impressed by the performance and appreciated our use of existing benchmarks in literature.

**Weaknesses**

One commonly noted weakness is the increased finetuning runtime of our proposed architecture, Attic. We acknowledge this concern. In our rebuttal, we argued that achieving higher accuracy should be the primary focus initially, as future work can address runtime optimizations using the numerous efficiency techniques developed for large language models. Applying these techniques is more effective after establishing a high-performing model, which is why we believe our Attic architecture is an important contribution.

The second point raised was the lack of certain implementation details in the paper. Specifically, two reviewers requested more information on how we implemented the TabZilla benchmark, including details on the train/validation/test splits, hyperparameter settings, and dataset selection from the benchmark. We provided all these details during the rebuttal, which seemed to resolve the concerns.

Once again, we appreciate your time and feedback. Your insights have been invaluable in refining our work.

---

### Public Comment · ~Xunye_Tian1 · 2025-08-04
**Open-source code link**

Can I know the open-source code link for this work?
Thanks.

---

> ### Public Comment · ~Felix_den_Breejen1 · 2025-08-11
>
> We don't have an open-source code link.
> Quickly after the ICLR decision, TabPFNv2 came out which has the exact same architecture as us.
> Also, during the ICML rebuttal, we discovered a bug in the bfloat16 TabPFN baseline, and we lost access to our GPUs so we could not fix the paper.
> In the end, we decided not to put our paper on arxiv and to not give the code link for these reasons.
> If you are interested in the code of the architecture, please use the TabPFNv2 repo or dig in our code attached in the supplementary materials, as that code should fully work.

---

### Meta-Review · Area_Chair_Zgam · 2024-12-22

**Metareview:**

This paper introduces Attic, a new architecture for in-context learning on tabular data, replacing the embedding per data point that Tab(Forest)PFN uses by an embedding for every data point and feature. The reviewers agree that the modification makes intuitive sense; the main strength is that results are strong compared to established baselines, on common benchmarks. The reviewers criticize the lack of theoretical justification, complexity (memory and runtime), unresolved instabilities in mixed-precision training, and fairness of evaluation due to the longer training time of Attic. Due to a combination of these issues, I am leaning towards rejection of the current version. The authors have improved the paper over the course of the rebuttal, and I encourage them to resubmit their work to the next venue.

**Additional Comments On Reviewer Discussion:**

The discussions focussed on the criticisms above, especially complexity (memory and runtime), unresolved instabilities in mixed-precision training, and fairness of evaluation due to the longer training time of Attic. The authors addressed questions about the experimental setup with TabZilla and added runtime numbers to the appendix, but they did not provide an apples-to-apples comparisons where all baselines get the same compute time, and could not resolve the stability issues in mixed-precision training question.

---

### Decision · Program_Chairs · 2025-01-22

Reject